# Importance of miRNA stability and alternative primary miRNA isoforms in gene regulation during *Drosophila* development

Li Zhou[1,2†], Mandy Yu Theng Lim[1,3†], Prameet Kaur[4], Abil Saj[5], Diane Bortolamiol-Becet[6‡], Vikneswaran Gopal[7], Nicholas Tolwinski[2,4], Greg Tucker-Kellogg[2], Katsutomo Okamura[1,3]*

[1]Temasek Life Sciences Laboratory, Singapore, Singapore; [2]Department of Biological Sciences, Faculty of Science, National University of Singapore, Singapore, Singapore; [3]School of Biological Sciences, Nanyang Technological University, Singapore, Singapore; [4]Division of Science, Yale-NUS College, Singapore, Singapore; [5]Cancer Therapeutics and Stratified Oncology, Genome Institute of Singapore, Singapore, Singapore; [6]Department of Developmental Biology, Sloan-Kettering Institute, New York, United States; [7]Department of Statistics and Applied Probability, Faculty of Science, National University of Singapore, Singapore, Singapore

*For correspondence:
okamurak@tll.org.sg

[†]These authors contributed equally to this work

Present address: [‡]CNRS Architecture et Réactivité de l'ARN, Université de Strasbourg, Strasbourg, France

Competing interests: The authors declare that no competing interests exist.

**Abstract** Mature microRNAs (miRNAs) are processed from primary transcripts (pri-miRNAs), and their expression is controlled at transcriptional and post-transcriptional levels. However, how regulation at multiple levels achieves precise control remains elusive. Using published and new datasets, we profile a time course of mature and pri-miRNAs in *Drosophila* embryos and reveal the dynamics of miRNA production and degradation as well as dynamic changes in pri-miRNA isoform selection. We found that 5' nucleotides influence stability of mature miRNAs. Furthermore, distinct half-lives of miRNAs from the *mir-309* cluster shape their temporal expression patterns, and the importance of rapid degradation of the miRNAs in gene regulation is detected as distinct evolutionary signatures at the target sites in the transcriptome. Finally, we show that rapid degradation of miR-3/–309 may be important for regulation of the planar cell polarity pathway component Vang. Altogether, the results suggest that complex mechanisms regulate miRNA expression to support normal development.
DOI: https://doi.org/10.7554/eLife.38389.001

## Introduction

Gene expression is regulated at multiple levels involving transcriptional and post-transcriptional mechanisms. Small RNAs including microRNAs (miRNAs) play central roles in post-transcriptional regulation of protein-coding genes. Typically, miRNAs are transcribed as primary transcripts (pri-miRNAs) by RNA polymerase II and their molecular architectures resemble those of mRNAs (*Kim et al., 2009*). The majority of miRNAs are processed from pri-miRNAs by a two-step cleavage mechanism involving two RNase III-class enzymes, Drosha and Dicer (*Okamura, 2012*). The resulting 21-23nt miRNA duplexes are loaded onto Argonaute family proteins and subsequently unwound to form mature effector complexes that regulate expression of target mRNAs.

Analogous to regulation of protein-coding genes, expression of miRNAs is regulated at both transcriptional and post-transcriptional levels (*Ha and Kim, 2014*). Transcription of pri-miRNAs is

**eLife digest** Cells produce proteins by feeding molecules that contain temporary copies of the gene for that protein through a complex structure called a ribosome. The ribosome follows the coded instructions in these molecules to build the protein. These temporary copies of the code are reusable. Even if the cell stops copying a gene it will continue to produce the protein for a short time.

MicroRNAs (often shortened to just miRNAs) can switch protein production off. These are short molecules that stick to the code of the protein-producing molecules. This renders the molecules unreadable to ribosomes, and also makes it a target for destruction by enzymes. Different miRNAs have different targets, helping to fine-tune the timing and amount of protein production.

There are two stages to the production of miRNAs. First, the cell copies the gene into primary transcripts (pri-miRNAs). Then, it turns these molecules into mature miRNAs. The cell can vary the number of pri-miRNAs made, control how and when they mature, and change the lifespan of the mature miRNAs. But, it is unclear how these processes all work together to achieve fine control of protein production.

Recent studies have revealed when, where and how much miRNA is present in developing organisms. So, scientists are now at the point where they can start to understand how cells control miRNA levels. Here, Zhou, Lim et al. created small RNA libraries at eight time windows during the development of fruit fly embryos. The libraries contained the mature miRNAs present at each developmental stage. Fruit fly embryos develop quickly, taking only 24 hours to make a larva from a single fertilized egg, and its genes must respond quickly.

When combined with existing datasets, the new data revealed how mature and pri-miRNAs change as fly embryos develop. Many miRNA genes sit close together, forming clusters in the fruit fly genome. Yet rather than make them all at once, the fly embryos often copied them in sets. So, as development progressed, different groups of miRNAs came into use. To achieve this, the cells copied different parts of the cluster at different times, and altered the way they processed the pri-miRNAs. The miRNAs from the same cluster lasted for different lengths of time, and the cells rapidly destroyed unwanted mature miRNAs. Together, these mechanisms shaped the timing and composition of each distinct set of miRNAs.

Understanding the control of miRNAs is an essential step in understanding how the cell regulates its genes. There are thousands of miRNA genes in the human genome, and a failure to control them can contribute to human diseases, including cancer. Future studies could extend this work by sampling other tissues of the fly, or tissues of other organisms, including humans.
DOI: https://doi.org/10.7554/eLife.38389.002

developmentally regulated as shown by in situ hybridization and transcriptome analysis in *Drosophila* (*Aboobaker et al., 2005*; *Graveley et al., 2011*; *Brown et al., 2014*; *Liu et al., 2017*). On the other hand, recent studies also revealed a variety of mechanisms post-transcriptionally regulating miRNA processing in a gene-specific or global manner. Sequence-specific RNA binding proteins can positively or negatively regulate miRNA processing through binding to pri- or pre-miRNAs (*Newman et al., 2008*; *Rybak et al., 2008*; *Heo et al., 2008*; *Viswanathan et al., 2008*; *Davis et al., 2008*; *Suzuki et al., 2009*; *Trabucchi et al., 2009*; *Guil and Cáceres, 2007*; *Treiber et al., 2017*). Furthermore, global miRNA processing activity can be altered depending on the expression level of the core miRNA processing factors and/or the status of post-translational modifications of miRNA processing factors (*Tang et al., 2010*; *Tang et al., 2013*; *Paroo et al., 2009*; *Herbert et al., 2013*; *Wan et al., 2013*; *Wada et al., 2012*). miRNAs are also generally destabilized in neuronal cells (*Krol et al., 2010*). As expected for important gene regulators, dysregulation of miRNA activity is often implicated in diseases. For example, aberrant expression levels of miRNA processing factors are observed in various cancers, and proteins that are involved in neurological disorders often have miRNA-related functions (*Adams et al., 2014*; *Emde and Hornstein, 2014*).

Although individual post-transcriptional mechanisms that regulate miRNA processing have been studied under some cellular conditions, the extent to which these mechanisms contribute to miRNA

expression profiles during natural development is not understood (*Rüegger and Großhans, 2012*). In addition, expression of alternative pri-miRNA isoforms was shown to be important for differential regulation of individual members of clustered miRNAs using cell lines, but its biological significance remains unknown (*Chang et al., 2015*; *de Rie et al., 2017*). A previous study performed genome-wide analysis using a panel of small RNA libraries focusing on clustered miRNAs to detect effects of post-transcriptional regulation with a limited time resolution led to identification of candidate genes that are post-transcriptionally regulated (*Ryazansky et al., 2011*).

Here, we use fly embryogenesis as a model system to study regulation of miRNA biogenesis during development. We generated small RNA libraries from 8- time windows that cover the entire fly embryogenesis and quantified mature miRNA levels. Using integrated analysis of this set of small RNA libraries and total RNA-seq libraries published by modENCODE (*Westholm et al., 2014*; *Duff et al., 2015*), we found that mature miRNA expression changes can be predicted relatively precisely based on the transcriptional activity after taking degradation of mature miRNAs into account. Our results suggested that processing efficiency and mature miRNA half-lives stay generally constant throughout embryogenesis except for very early stages. However, some individual miRNAs show distinct half-lives, and miRNA stability plays a significant role in shaping the miRNA expression profile and in turn influences evolution of target sites in 3'UTRs. Our results also indicate that transcriptional start/termination sites (TSSs/TTSs) are flexibly used to express distinct sets of miRNAs from a single cluster and selection of alternative pri-miRNA isoforms is regulated in a developmental stage and signaling dependent manner.

These results provide insight into miRNA regulation in developmental processes at the global level and reveal complex mechanisms that support precise regulation of miRNA expression.

## Results

### Global changes in the bulk miRNA abundance during embryogenesis

To understand how the miRNA expression profile changes during fly embryogenesis, we prepared small RNA libraries in 2 hr windows from 0 to 12 hr after egg laying (hAEL) and 6 hr windows from 12-24hAEL. We prepared libraries in biological triplicate, each of which contained 9.8–23.8 million reads perfectly mapping to the *Drosophila melanogaster* genome (*Supplementary file 1* Sheet 1). To study the composition of the small RNA population, small RNA reads were first grouped under four categories: (1) miRNA, (2) endo-siRNA + piRNA, (3) reads from abundant non-coding RNAs (rRNAs and tRNAs etc.) and (4) other (*Figure 1A*; *Supplementary file 1* Sheet 2). In 0-2hAEL embryos, category two small RNAs constituted >50% of the small RNA population, consistent with the large amount of maternally deposited piwi-interacting RNAs (piRNAs) produced from various transposons (*Brennecke et al., 2008*). The fraction of category two small RNAs continuously declined until 12-18hAEL, with a concomitant increase of the miRNA fraction. In 18-24hAEL, we observed a reproducible increase of small RNAs derived from abundant ncRNAs. Closer inspection revealed that these small RNAs were mainly derived from rRNAs, and no specific size peaks were observed with this category of small RNAs in this time window (*Figure 1—figure supplement 1*). These results suggested that rRNA-derived small RNAs may be degradation intermediates with no functionality, but the reason for this rapid increase of these fragments in this developmental stage is not known.

To account for the change in the total small RNA population size, we used external standards for read count normalization (*Locati et al., 2015*). The levels of small RNA species were estimated by dividing the read counts by the number of reads matching to the spike-in oligo sequences introduced to the RNA sample (see Materials and methods; *Supplementary file 1*, Sheet 3). The normalized values were expressed as reads per thousand spike-in reads (RPTS). We believe that this normalization scheme permits more accurate comparisons of the miRNA levels in different libraries as compared to conventional normalization methods. The conventional approach using the total counts of mappable reads assumes that the total amount of small RNAs stays constant in different samples, but such assumptions might not be appropriate for certain analyses. For example, the large number of rRNA reads in 18-24hAEL (*Figure 1A*; *Figure 1—figure supplement 1*) could strongly affect normalization factors of the conventional methods.

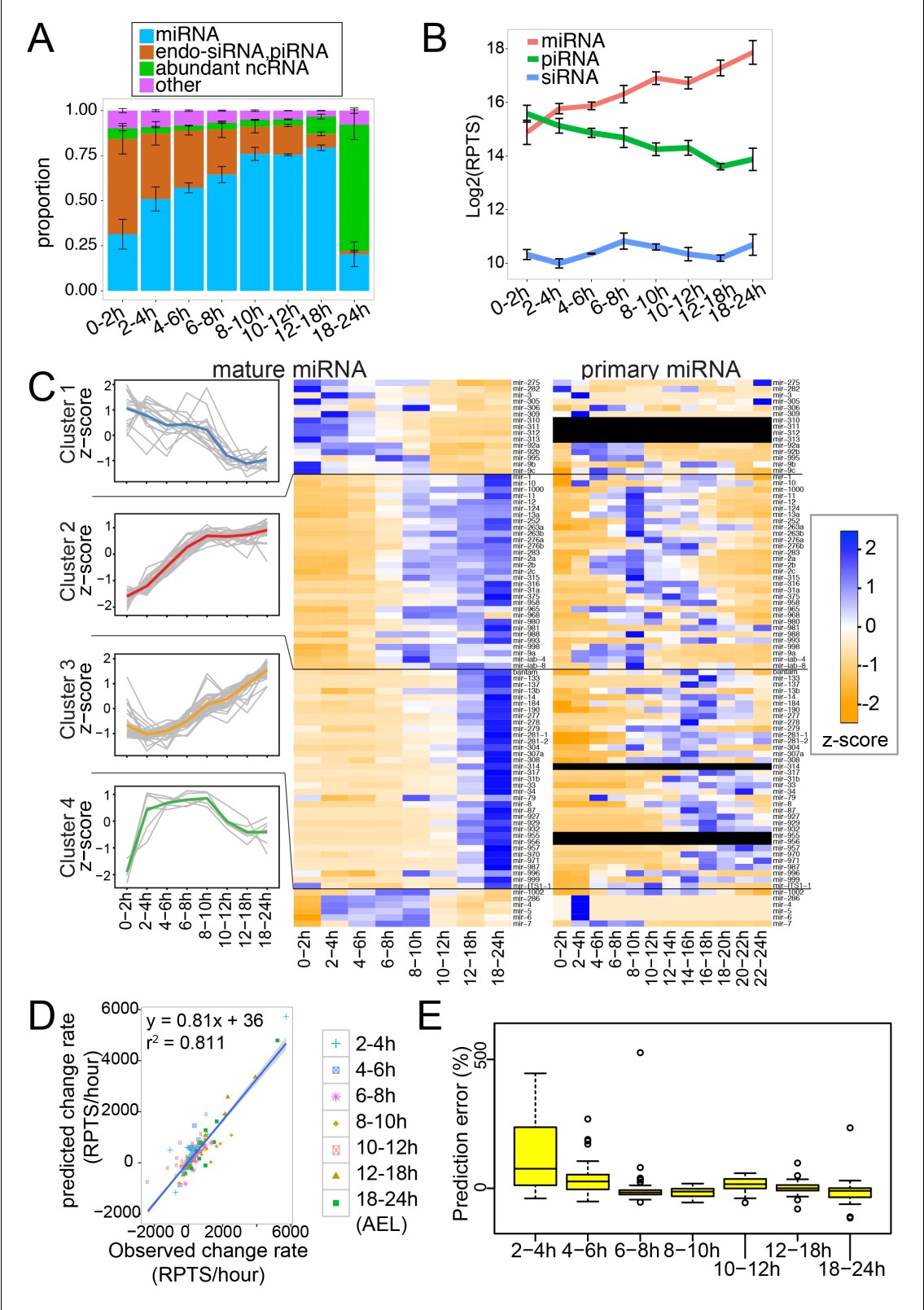

**Figure 1.** Mature and pri-miRNA profiles revealed by RNA-seq analysis. (A) Composition of small RNA libraries. Small RNA reads were grouped into four categories, and their fractions in the library were plotted. The average and standard error of mean of three replicate libraries are shown. (B) Changes in the global levels of small RNA classes. The averages of total read counts of miRNAs (red), piRNAs (green) and siRNAs (blue) were plotted. Error bars indicate standard error of mean. Small RNA read counts were normalized against the spike-in count of each library and expressed as reads

*Figure 1 continued on next page*

*Figure 1 continued*

per thousand spike-in reads (RPTS). (C) miRNA genes were grouped into four clusters by k-means clustering (k = 4) and expression z-scores of individual miRNAs are shown in the left panel (gray lines). The average z-score for each cluster is shown as a colored line. Expression profiles of mature miRNAs (middle) or pri-miRNAs (right) in staged fly embryos are shown in the heatmap. The heatmap was color-coded according to the z-score calculated across the 8 or 12 time-windows for each miRNA gene. The pri-miRNA level was determined as the read density in the 100nt window immediately upstream of the miRNA hairpin. (D) The mature miRNA change rates predicted by multiple regression analysis were plotted against observed values. (E) Error rates estimation for each time window. Error rates were determined by the following formula. Error rates = (predicted - observed miRNA change rate) * (window size) * 100 / (mature miRNA level). Distributions of prediction error rates were plotted. Four outliers in the 2–4 hr time window (1402.5, 972.8, 954.3 and 739.9%) are not shown. Genes with at least 10 RPTS in the time window were used for this analysis.
DOI: https://doi.org/10.7554/eLife.38389.003

The following figure supplements are available for figure 1:

**Figure supplement 1.** Size distribution of small RNA reads mapping to abundant ncRNAs.
DOI: https://doi.org/10.7554/eLife.38389.004
**Figure supplement 2.** mRNA levels of miRNA/piRNA processing factors.
DOI: https://doi.org/10.7554/eLife.38389.005
**Figure supplement 3.** Relative levels of total RNA and AGO1 protein per embryo in developing embryos.
DOI: https://doi.org/10.7554/eLife.38389.006

We first analyzed the overall abundance of miRNAs after normalizing read counts against the spike-in counts (*Figure 1B*; *Supplementary file 1* Sheet 2). We observed a continuous increase of the total mature miRNA abundance throughout embryogenesis. This may be consistent with the notion that early embryos generally show relatively low miRNA activity and differentiated cells rely more strongly on miRNA-mediated mechanisms for gene regulation (*Lu et al., 2005*; *Kumar et al., 2007*; *Kumar et al., 2009*; *Suh et al., 2010*; *Ohnishi et al., 2010*). On the other hand, we observed a gradual decrease of the piRNA population (*Figure 1B*; *Supplementary file 1* Sheet 2). This was also consistent with expression patterns of mRNAs encoding the piRNA pathway components with high levels in early embryos (*Figure 1—figure supplement 2*) and largely restricted to gonads in late stages (*Ishizu et al., 2012*).

We analyzed individual miRNA levels in more detail (*Figure 1C*, middle). To test the accuracy of our library-based miRNA profiling, we performed Northern blotting analysis for seven randomly chosen miRNAs (*Figure 2*). The normalized miRNA expression values estimated by library analysis generally showed good agreement with our Northern blotting results (*Figure 2*, *Figure 2—figure supplement 1* and *Supplementary file 2*), indicating that our sequencing analysis accurately estimated levels of miRNA expression relative to the amount of total RNA. We found that the amount of total RNA per embryo stayed relatively constant throughout embryogenesis (*Figure 1—figure supplement 3A*). Therefore, we believe that our library analysis results using spike-in normalization reflect the relative abundance of each miRNA species per embryo (*Figure 1B*).

Having confirmed the accuracy of normalized values, we further analyzed changes of individual miRNA levels. For each of the 87 miRNAs whose expression levels satisfied our cutoff (>50 RPTS), we calculated relative expression levels in the eight time windows, and grouped the miRNA genes into four clusters using the k-means clustering method (*Hartigan and Wong, 1979*) (*Figure 1C*, left). Cluster 1 included 15 miRNAs that showed the highest expression levels in early stages. The majority of miRNA genes in this group were maternally deposited miRNAs with weak zygotic expression (*Lee et al., 2014*; *Marco, 2015*). The rapid decrease of maternal mature miRNA species was consistent with Wispy-mediated degradation of maternal miRNAs as reported previously (*Lee et al., 2014*). Cluster 4 included 6 genes and showed detectable decrease in late stages. This observation suggested that these miRNAs are relatively unstable, and the small number of genes in this category was consistent with the conclusion in previous cell culture-based studies that a small subset of miRNAs is degraded relatively quickly (*Duffy et al., 2015*; *Marzi et al., 2016*).

The majority (75.8%; 66 out of 87 genes) of genes belonged to clusters 2 and 3 that showed increasing trends throughout embryogenesis. This indicated that the increase in the expression level is a general trend for many miRNA genes in fly embryos, and that the increase in the bulk miRNA abundance was not caused by a small number of miRNA species. Coincidentally, the AGO1 protein level was also lower in very early embryos (*Figure 1—figure supplements 3B*, 0–2h and 2–4h lanes). This may suggest that the AGO1 protein level may be a limiting factor that determines the bulk

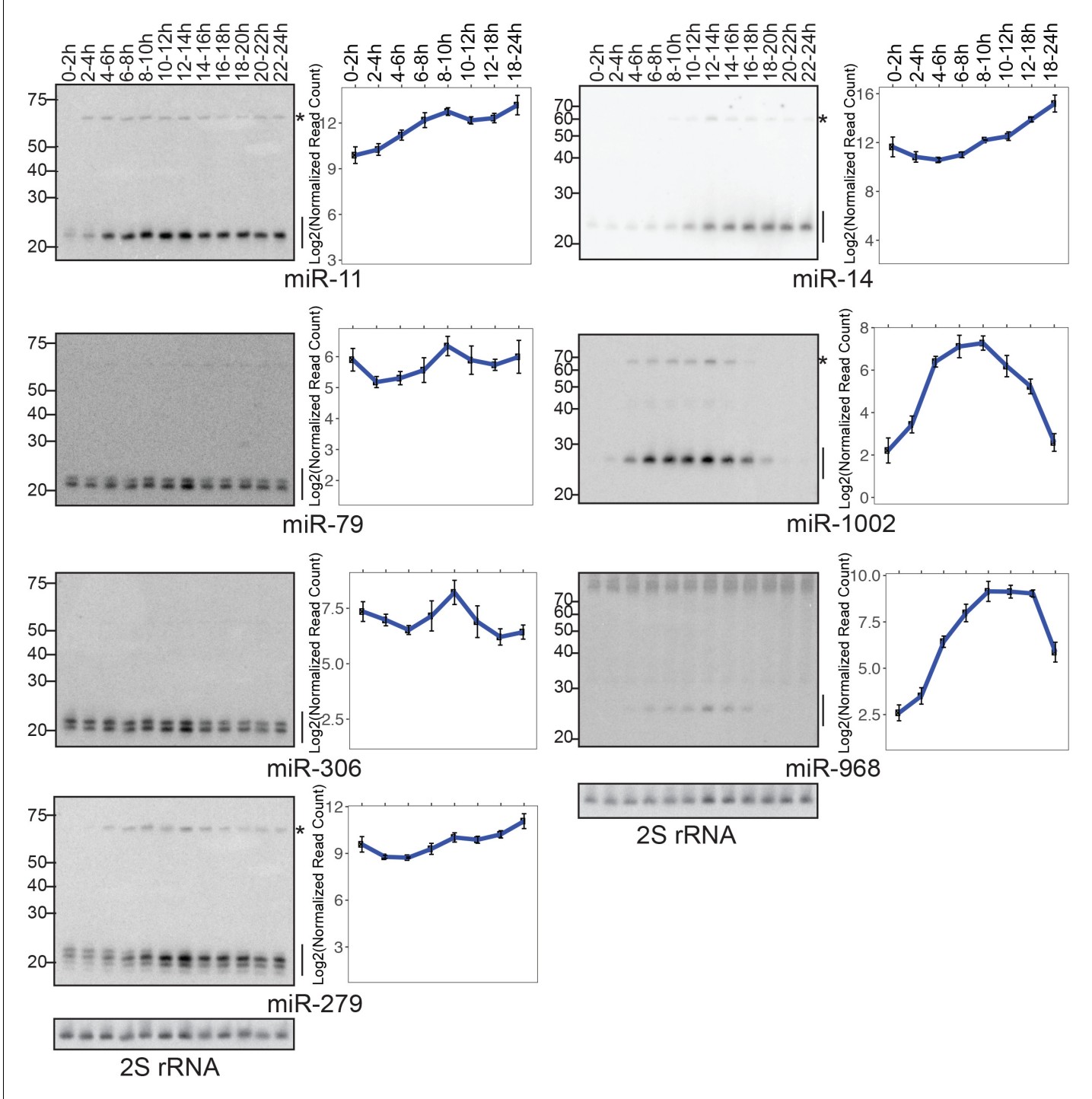

**Figure 2.** Verification of expression data by Northern blotting. (Left panels) RNA samples from staged embryos were resolved by 15% denaturing PAGE and probed for the mature species of the indicated miRNA gene. Mature miRNA and precursor species are indicated by lines and asterisks respectively. (Right panels) Mature miRNA quantification using small RNA library data. Average values of triplicates are shown. Error bars indicate the standard error of mean.

DOI: https://doi.org/10.7554/eLife.38389.007

The following figure supplement is available for figure 2:

**Figure supplement 1.** Quantification of *Figure 2* Northern blotting results.
DOI: https://doi.org/10.7554/eLife.38389.008

miRNA abundance. Alternatively, the lower level of mature miRNAs may trigger degradation of apo-AGO1, as observed in tissues depleted of miRNA processing factors (*Smibert et al., 2013*).

These results confirmed distinct behaviors of small RNA families during fly embryogenesis, reflecting their distinct biological functions. The large difference of the total miRNA levels (eight times increase from 0-2hAEL to 18-24hAEL) estimated by our method underscores the importance of the selection of normalization methods, as the conventional RPM (reads per million) normalization method generally disregards the change in the size of small RNA population.

## Transcription levels of individual pri-miRNAs estimated by total RNA-seq analysis

To profile transcription levels of individual miRNA genes on a genome-wide scale, we reanalyzed ribosomal RNA-depleted, stranded total RNA-seq data from fly embryos on a similar time course that were generated by the modENCODE consortium (*Westholm et al., 2014*; *Duff et al., 2015*). The expression level of each miRNA gene was estimated using the density of total RNA-seq reads in the 100-nucleotide window immediately upstream of the pre-miRNA hairpin (*Figure 1C*, right; *Supplementary file 3*).

Pri-miRNA species (or introns in the case of miRNAs residing in introns of protein-coding mRNAs) are believed to have much shorter half-lives compared to mature mRNAs or mature miRNAs (*Gaidatzis et al., 2015*; *Chang et al., 2015*; *Nojima et al., 2015*). Consistent with this notion, transient peaks of the pri-miRNA levels were often observed, suggesting that miRNA transcription is activated generally in short time-windows (*Figure 1C*, right). For example, even for the miRNAs in clusters 2 and 3 that showed the highest mature miRNA levels in the last time window, their corresponding primary miRNAs generally exhibited expression peaks in earlier time windows while mature miRNA remained present for at least several hours.

The change rate of mature miRNA level per unit of time (hereafter 'miRNA change rate') should be determined by two factors: new synthesis and degradation. We sought to test how well the miRNA change rate could be predicted by taking only these two factors into consideration. For simplicity, we approximated the model based on the following two assumptions: the level of transcription would be the primary determinant of the miRNA synthesis rate and the amount of mature miRNA that is degraded at a given moment would be proportional to the amount of miRNA products present at the moment. Based on these assumptions, we performed multiple linear regression analysis to obtain coefficients by fitting the quantified values of mature and primary miRNAs to the following equation: $z \sim ax + by + c$, where $z$ = change rate of mature miRNA, $x$ = initial miRNA level, and $y$ = upstream density (*Supplementary file 4*, see Materials and methods for details). When we tested the correlation between the predicted and observed change rates for all time windows of all miRNAs, we found that the change rates could be generally accurately predicted ($r^2$ = 0.81; *Figure 1D*). This suggested that, for the majority of miRNAs, the production rate per pri-miRNA molecule as well as the degradation rate of mature miRNAs were relatively constant throughout embryogenesis. Nevertheless, when we analyzed the prediction accuracy in each window, we observed a trend where the predicted change rate tended to be higher than the observed data in early time windows (*Figure 1E*). This may mean that mature miRNAs were less efficiently processed from pri-miRNAs in early embryos. It is also possible that pri-miRNA degradation might be slower in these windows, leading to an overestimation of transcriptional activity. Although we excluded miRNAs that are deposited as maternal miRNAs at high levels from this analysis, the small amount of maternal miRNAs and their rapid degradation (*Lee et al., 2014*) might also contribute to this inaccurate prediction in this window. In other time windows, the distribution of prediction errors did not show clear trends, suggesting that the kinetics of miRNA production/degradation stayed relatively constant in these windows (*Figure 1E*).

Our integrated analysis of mature and primary miRNAs confirmed the general assumption that transcription level is important for determining the expression patterns of mature miRNAs and suggested that global miRNA production and degradation rates are relatively constant during embryogenesis except for early embryos. However, we note that individual predicted values may not be accurate due to the inaccuracy of sequence-based quantification and the relatively low time-resolution of our time course. In addition, it is also possible that there are a small number of individual miRNAs that are regulated post-transcriptionally in a time-window specific manner in embryos.

## Influences of the 5' nucleotides on mature miRNA stability

While production of miRNAs would be the major determinant of miRNA expression profiles, degradation of mature miRNAs should also play a role. The identity of 5' end nucleotide of the guide RNA influences the interactions between the guide RNA and the MID domain of Argonaute with 5'-uridine (5'-U) showing the highest affinity among the four nucleotides (*Frank et al., 2010*). Since mature miRNAs are believed to be protected by Argonaute proteins from degradation, we wondered if the identity of the 5' nucleotide could also influence the stability of miRNAs.

We asked whether our results of multiple linear regression analysis (*Supplementary file 4*) could be used to estimate miRNA degradation rates. Our analysis relies on the assumption that the mature miRNA change rate is determined by the miRNA production rate that is proportional to the level of transcription and the miRNA degradation rate that is proportional to the level of the mature miRNA abundance. In theory, the slopes associated with the mature miRNA abundance values should be negative because we expect the values to reflect the degradation rates per mature miRNA molecule per unit of time (Column 'Coef_mature_Level' in *Supplementary file 4*), while the coefficients for pri-miRNA abundance (Column 'Coef_Updensity' in *Supplementary file 4*) were expected to be positive values because they would reflect processing efficiency. However, we found that some individual miRNAs showed poor correlation (Column 'r2' in *Supplementary file 4*; 12 out of 45 genes showed $r^2 < 0.5$) when data points were fitted to a linear plane. In addition, many of the coefficients for the mature miRNA levels were unexpectedly positive, suggesting that there were large errors in the estimated coefficients. Nevertheless, when the values were compared between groups of miR-NAs with 5'-U and other 5'-nucleotides, we observed a trend whereby the coefficient values associated with the mature miRNA abundance were generally higher with mature miRNAs carrying 5'-U than those carrying other 5'-nucleotides (p=*0.01*, one-tailed Kolmogorov-Smirnov test; mean$_{5'U}$=-0.0016, mean$_{non-5'U}$=-0.088; median$_{5'U}$=0.013, median$_{non-5'U}$= -0.087) (*Figure 3—figure supplement 1*). One may expect this trend if miRNAs with 5'-U are more stable than miRNAs with other nucleotides at the 5' end. However, due to the observed inaccuracy in estimating relative degradation rates by our genome-wide analysis, which even resulted in the unexpected positive median of the coefficient values for 5'-U species, we were unable to make a confident conclusion.

Inspired by the observations above, we experimentally tested the hypothesis that 5' nucleotides influence miRNA stability by mutating a mature miRNA carrying a non-5'-U. For this test, we chose miR-283, which has an adenine (A) at the 5' end, as a model miRNA and generated a construct to express a miR-283 mutant carrying a 5'-U (A-to-U mutant). The strand selection of this miRNA is highly asymmetric with a strong bias for 5p accumulation, and introducing a 5'-U to the 5p-5' end is not expected to change the strand selection (*Kozomara and Griffiths-Jones, 2014*). The wild-type and mutant plasmids were transfected to S2-R+ cells and expression was induced by CuSO$_4$ for 24 hr, followed by the termination of induction by replacing the medium with a new medium containing the CuSO$_4$ chelator (*Djuranovic et al., 2012*). RNA was extracted on a time course after removal of the inducer, and the reduction kinetics of the mature miR-283 species was analyzed by Northern blotting (*Figure 3B and C*). We observed a slight (~20%) but reproducible stabilization of the mutated miR-283 with average half-lives of 14.9 and 18.3 hr for wild-type and A-to-U mutant, respectively (p<0.02, paired one-tailed t-test).

It was formally possible that our experiments might be measuring half-lives of a mixed population containing miRNA duplexes before loading to Argonautes and mature miRNAs loaded in Argonaute complexes. However, the miR-283 star strand was almost undetectable even in the first time window after the termination of *mir-283* expression from the plasmid (*Figure 3B*). Given that unwinding of miRNA duplex occurs during loading to Argonautes (*Kawamata and Tomari, 2010*), this result excluded the possible contribution of diced miRNA strands in the duplex form before Argonaute loading to our half-life measurements.

To test whether this is specific to miR-283 and whether other nucleotides have effects on mature miRNA stability, we generated additional mutants (*Figure 3D*, *Figure 3—figure supplement 2*). When we changed the 5'-As of two other miRNAs to 5'-Us, we observed stabilizing effects similar to the miR-283 A-to-U mutant. On the other hand, mutating 5'-A to 5'-G did not significantly affect the stability. We also observed significant stabilization when the mature miRNA 5' ends were changed to 5'-C, which is the preferred 5' nucleotide by the *Drosophila* siRNA Argonaute AGO2 (*Figure 3D*, *Figure 3—figure supplement 2*) (*Ghildiyal et al., 2010*; *Czech et al., 2009*).

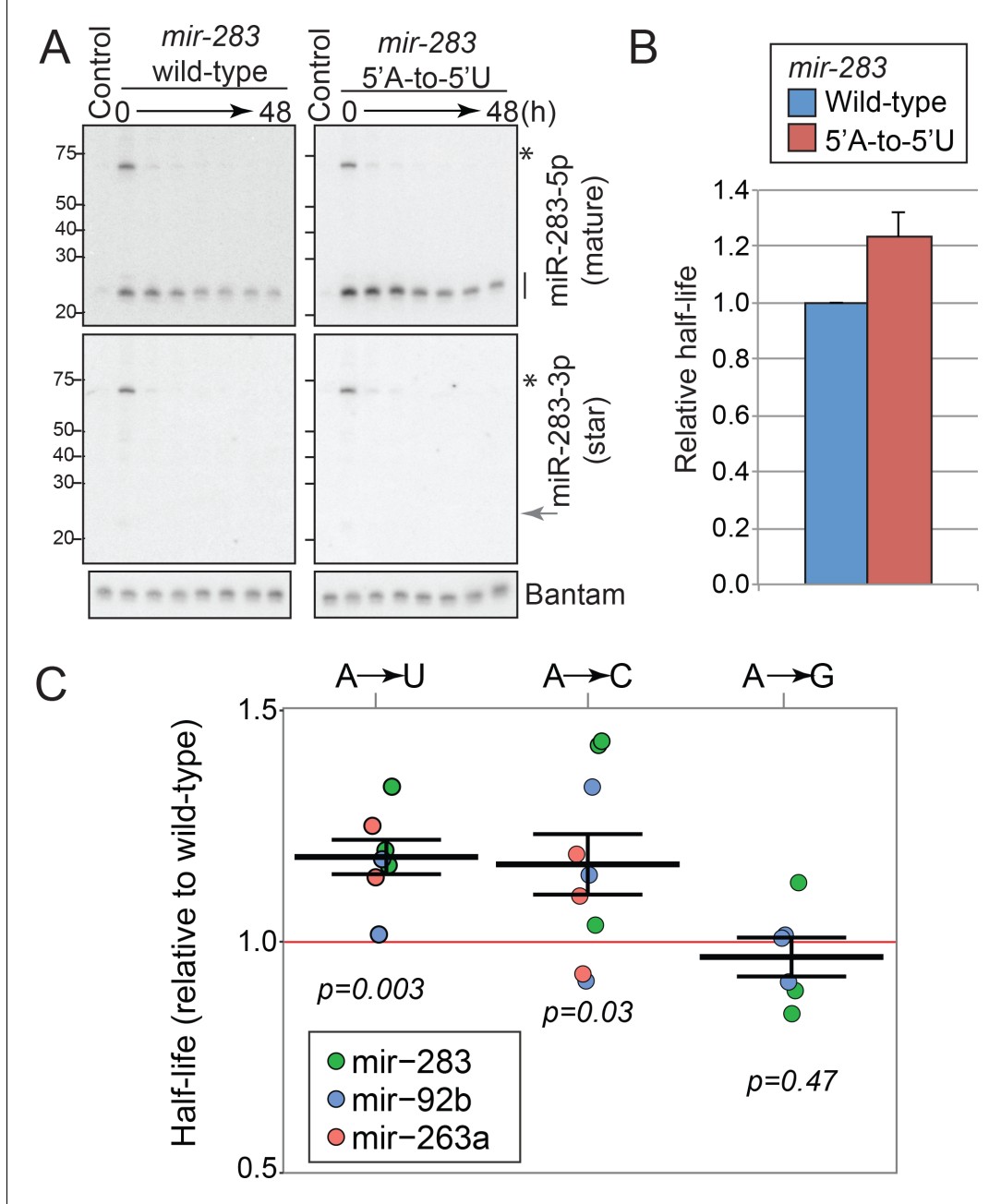

**Figure 3.** Effects of the 5′ nucleotide on miRNA stability. (**A**) Experimental validation of the role for miRNA 5′ nucleotides in mature miRNA stability. A genomic DNA fragment containing *mir-283*-cluster locus was cloned in a $CuSO_4$-inducible plasmid (*mir-283* wild-type). The 5′-nucleotide mutation was introduced by site-directed mutagenesis to generate the mutant plasmid (*mir-283* A-to-U) and the plasmids were used for transfection of S2-R+ cells. Transfected cells were cultured for 24 hr in the presence of $CuSO_4$ to induce the expression, and the expression was terminated by replacing with medium containing the $CuSO_4$ chelator (time 0). The time intervals after the withdrawal of the inducer are: 0, 6, 12, 24, 30, 36 and 48 hr. For control lanes, cells were transfected with the same plasmid but $CuSO_4$ induction was omitted. RNA samples were prepared for Northern blotting analysis at the indicated time points to monitor the reduction rates of mature miRNA species. Asterisks and lines indicate the positions of precursor and mature miR-283 signals, respectively. A representative figure of three attempts is shown. The miR-283 star strand was present at very low levels compared to the mature strand, confirming that the miRNA stability measurements reflect the half-lives of mature miRNAs that are already loaded to the Argonaute complex. Endogenous bantam was detected and used as a loading control. (**B**) Quantification of the triplicates of miR-283 mutant analysis shown in (**A**). The relative miRNA half-life for miR-283 (**A–to–U**) was normalized by that from the corresponding wild-type result in the same replicate. The average and the standard deviation are shown. The miR-283 A-to-U mutant exhibited a slightly extended half-life (p=0.018, one-tailed paired t-test, N = 3). (**C**) Mature miRNA stability of mutant miRNAs. 5′ ends of the mature strands were mutated to indicated nucleotides and the time course experiment was done using S2-R+ cells expressing the wild-type or mutant miRNAs under the CuSO4-inducible promoter, similar to *Figure 6B and C*. The relative half-

*Figure 3 continued on next page*

*Figure 3 continued*

life of the mutant miRNA was determined for each replicate, and individual values (dots) and means ±standard errors (lines) are shown. The colors of dots indicate the miRNA backbone used for mutagenesis. Student's t-test p-values are shown in the chart. Representative images can be found in *Figure 6—figure supplement 2*. Raw data can be found in *Figure 3C—source data 1*.

DOI: https://doi.org/10.7554/eLife.38389.009

The following source data and figure supplements are available for figure 3:

**Source data 1.** Raw data for *Figure 3C*.

DOI: https://doi.org/10.7554/eLife.38389.012

**Figure supplement 1.** Distributions of estimated relative half-lives for 5'-U miRNAs and miRNAs with other 5' nucleotides.

DOI: https://doi.org/10.7554/eLife.38389.010

**Figure supplement 2.** Mutagenesis of 5' nucleotides in *mir-283*, *mir-92b* and *mir-263a* backbones.

DOI: https://doi.org/10.7554/eLife.38389.011

The results revealed the importance of 5' nucleotides in stabilizing mature miRNAs, in addition to their previously characterized roles in miRNA loading to Argonautes (*Meister, 2013*; *Frank et al., 2010*).

## Analysis of clustered miRNAs uncovers the complexity of miRNA regulation

miRNA genes often form clusters in the genome and those clustered miRNAs are believed to be co-transcribed as polycistronic units. Therefore, if the level of transcription is the major determinant of the miRNA expression levels, miRNAs within a cluster should show similar changes in their expression levels. Nevertheless, we observed several time windows where miRNA cluster members showed distinct expression patterns (*Figure 4*).

Overall, from the 21 miRNA clusters in the *D. melanogaster* genome (*Kozomara and Griffiths-Jones, 2014*), we could detect 52 mature miRNA species derived from 13 miRNA clusters after removing multi-copy genes (i.e. families with the same mature sequence) (*Figure 4*). As expected, the majority of mature miRNA species from a single cluster showed similar expression patterns (adjusted p>0.01, one-way ANOVA analysis). However, four clusters exhibited significant differences in at least one time window (*Figure 4*, highlighted by red rectangles, *Supplementary file 5*).

These results observed with clustered miRNAs suggested that a small number of miRNAs are regulated by additional mechanisms rather than simple regulation of transcriptional activity. In the following sections, we describe two potential mechanisms that may underlie the observed differences in expression changes between cluster members.

## Primary miRNA isoforms produce distinct sets of miRNAs from a miRNA cluster

We sought to understand the molecular basis of distinct expression patterns seen with the *mir-317* cluster miRNAs (*Figure 4*). This cluster consists of three miRNAs (miR-317, miR-34 and miR-277) and plays important roles during fly aging through multiple mechanisms including modulation of ecdysone signaling and branched-chain amino acid catabolism via the actions of miR-34 and miR-277, respectively (*Liu et al., 2012*; *Esslinger et al., 2013*; *Xiong et al., 2016*). However, previous studies reported seemingly inconsistent results on their expression levels, which showed up-regulation of miR-34 and down-regulation of miR-277 during aging (*Liu et al., 2012*; *Esslinger et al., 2013*). Our primary miRNA analysis suggested a potential mechanism that may explain the distinct expression patterns of these miRNAs.

We were able to identify five major primary miRNA isoforms with combinations of alternative TSSs and TTSs in this locus (*Figure 5A*). Interestingly, one of the isoforms only covered the *mir-277* hairpin, starting in downstream of the *mir-317* hairpin and ending in upstream of the *mir-34* hairpin. This short isoform was the main isoform expressed in embryos and its expression peak was seen at ~14–18 hAEL (*Figure 5—figure supplement 1A*), coinciding with the time window where the significant difference between miR-277 and miR-317/–34 expression changes was observed (*Figure 4*). Further analysis of total and small RNA-seq data in other tissues and developmental stages revealed a more dramatic difference in the relative levels of these miRNAs (*Figure 5A* and *Figure 5—figure supplement 1B*). In contrast to embryos, adult tissues use the long isoforms as the main primary

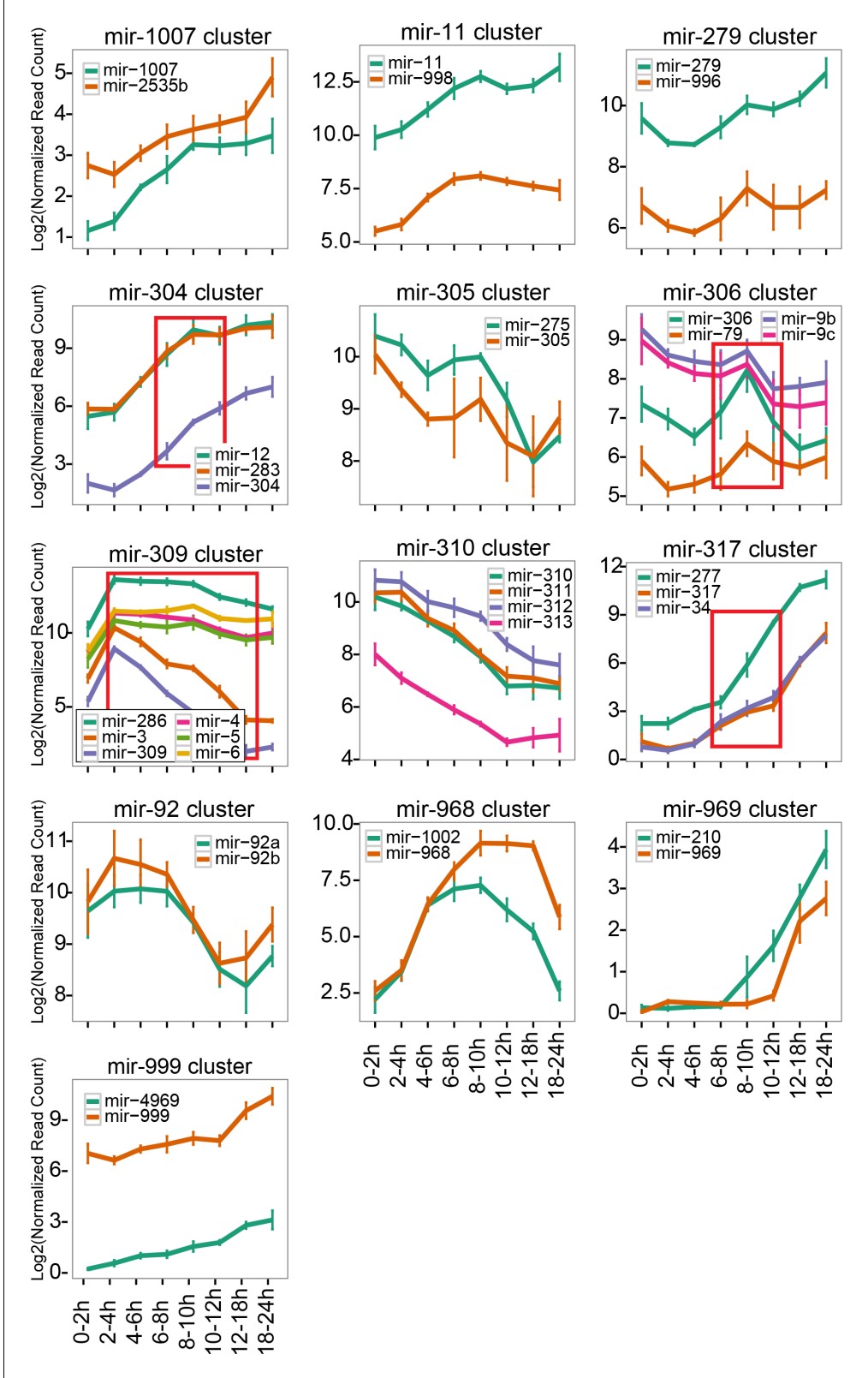

**Figure 4.** Expression changes of clustered miRNAs. Normalized read counts of individual miRNA clusters were plotted. The lines and error bars indicate the averages and standard error of mean respectively. The time windows showing significant expression changes are indicated by red rectangles (ANOVA p<0.01, N = 3, p-values are shown in *Supplementary file 5*).

DOI: https://doi.org/10.7554/eLife.38389.013

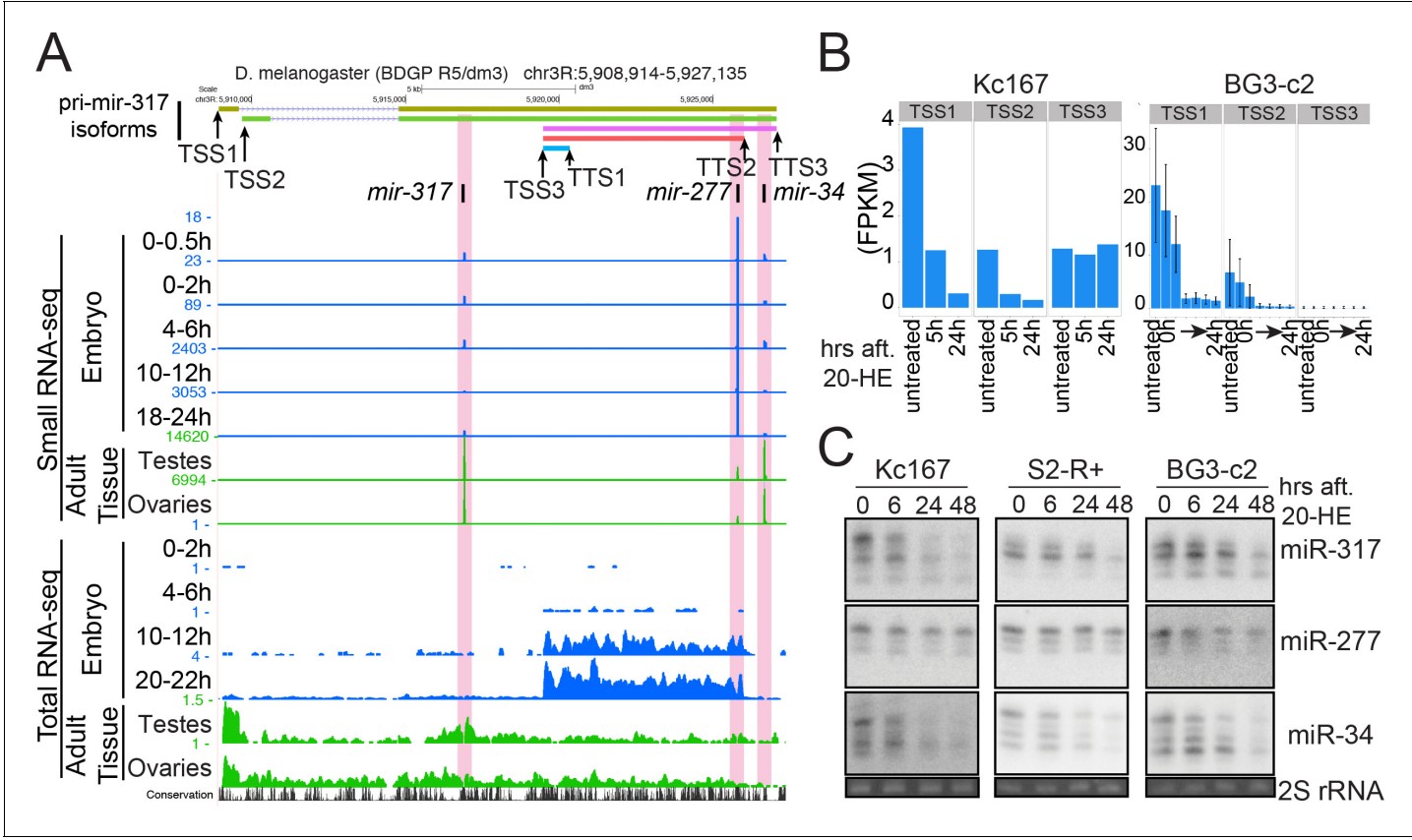

**Figure 5.** Distinct miRNA subsets are produced from the *mir-317* cluster by alternative pri-miRNA isoforms. (**A**) UCSC genome browser snapshot of the *mir-317* cluster locus. Small RNA-seq (upper) and total RNA-seq (lower) data are shown. In embryos (blue) miR-277 expression is higher than the other two miRNAs, and even higher in late embryos, which is consistent with the high expression of pri-miRNA isoform starting from TSS3 in embryos. In testes and ovaries, long isoforms starting from TSS1 and 2 are dominant, and the miR-277 level is lower than miR-317 and miR-34. In all tissues and embryonic stages, the ratios between miR-317 and miR-34 remain similar. (**B**) Reanalysis of published total RNA-seq data from cultured cells treated with ecdysone. The sum of FPKMs of the isoforms sharing the same TSS was plotted. (**C**) Northern blotting analysis of mature miR-317,–277 and −34 levels after 20-HE (hydroxyecdysone) addition in Kc167, S2-R+ and BG3-c2 cells. Cells were treated with 20-HE for the indicated time and total RNA was separated on a 15% denaturing acrylamide gel. miR-317 and miR-34 were decreased after 20-HE addition in Kc167 and S2-R+ cells, while the decrease of miR-277 was much weaker in these cell lines. In contrast, miR-277 was also decreased in BG3-c2 cells, in which the level of the *mir-277* specific short isoform was very low even in the absence of 20-HE (Panel B, TSS3). See **Figure 5—figure supplement 3** for the quantified results of tripricates.
DOI: https://doi.org/10.7554/eLife.38389.014

The following source data and figure supplements are available for figure 5:

**Figure supplement 1.** UCSC Genome Browser screen shot of the *mir-317* cluster locus with total RNA-seq tracks.
DOI: https://doi.org/10.7554/eLife.38389.015

**Figure supplement 2.** UCSC Genome Browser screenshot of the *mir-317* cluster locus with total RNA-seq and Histone H3K4me3-ChIP tracks.
DOI: https://doi.org/10.7554/eLife.38389.016

**Figure supplement 3.** Quantification of the Northern blotting results shown in **Figure 4C** and their biological replicates.
DOI: https://doi.org/10.7554/eLife.38389.017

**Figure supplement 3—source data 1.** Raw data for **Figure 5—figure supplement 3—source data 1**.
DOI: https://doi.org/10.7554/eLife.38389.018

*mir-317* isoform. Consistent with the long isoform expression, relative levels of miR-317/–34 compared to that of miR-277 in ovaries and testes were much higher than those in embryos (*Figure 5A*). The higher expression levels of miR-317 and −34 may imply that these two miRNA hairpins are more efficiently processed from pri-miRNAs than miR-277. Supporting the idea that the alternative pri-miRNA isoforms were derived from alternative TSSs, we observed consistent changes in the histone modification status at these sites (*Figure 5—figure supplement 2*). Although we do not formally exclude the possibility that the observed pri-miRNA isoforms may represent processing

intermediates as seen in previous studies (*Du et al., 2015*), we favor the hypothesis that the relative levels of *mir-317* cluster miRNAs could be altered by the selection of alternative TSS and TTS.

Expression of the *mir-317* cluster miRNAs is controlled by the insect steroid hormone ecdysone (*Sempere et al., 2003*; *Ameres et al., 2010*; *Xiong et al., 2016*). We were interested in testing if ecdysone has differential effects on individual *pri-mir-317/–277/−34* isoforms. In the embryonic Kc167 cells, we found that both long (starting at TSS1/2) and short (starting at TSS3) pri-miRNA isoforms were expressed in the absence of ecdysone and showed intermediate ratios of mature miR-277 and mature miR-34/–317 levels compared to embryos (high miR-277) and adult tissues (low miR-277) (*Figure 5A and B*). When Kc167 cells were treated with 20-hydroxyecdysone (20-HE), only the long transcripts starting at TSS1/2 were strongly down-regulated (*Figure 5B*). Consistent with the differential effects of ecdysone on the *pri-mir-317*-cluster isoforms, miR-317 and miR-34 decreased more strongly than miR-277 in Kc167 cells (*Figure 5C*, *Figure 5—figure supplement 3*). We observed a similar trend in another embryonic cell line, S2-R+ (*Figure 5C*, *Figure 5—figure supplement 3*). In contrast, BG3-c2, a cell line derived from larval nervous system (*Ui et al., 1994*), only expressed the long isoforms at detectable levels (*Figure 5B*). The three miRNAs decreased similarly in BG3-c2 cells when 20-HE was added to the medium (*Figure 4C*, *Figure 5—figure supplement 3*). These results using cell lines validated the principle that the transcriptional control through multiple promoters can influence the usage of TSS and TTS, resulting in differential regulation of individual miRNAs within a cluster.

These results provide a possible explanation for the distinct expression changes of the *mir-317* cluster miRNAs seen during aging (*Liu et al., 2012*; *Esslinger et al., 2013*). The flexible use of alternative TSSs/TTSs allows complex regulation of biological processes by up- or down-regulating subsets of miRNAs within individual clusters in response to stimuli, including hormones.

## Roles of miRNA degradation rates in shaping miRNA expression profiles

The *mir-309* cluster is another cluster whose members showed distinct expression patterns (*Figure 4*). This locus is transcribed at a very early embryonic stage, and its transcription ceases shortly after the activation of zygotic transcription (*Aboobaker et al., 2005*; *Graveley et al., 2011*). This cluster is essential for normal maternal-to-zygotic transition, although the exact roles of individual miRNAs in the cluster are not well understood (*Bushati et al., 2008*). The modENCODE RNA-seq library set confirmed the transient activation of transcription of this cluster in a very short time window (*Figure 5A*)(*Graveley et al., 2011*; *Aboobaker et al., 2005*). We were interested in looking at the changes in the expression levels of individual miRNAs in this cluster. We assumed that the decrease rate of the mature miRNA level after 2-4hAEL would reflect the degradation rate of each miRNA species, since there is no evidence for new miRNA synthesis from this cluster after this time window.

When we plotted expression levels of the *mir-309* cluster miRNAs in the eight time windows, we noticed that these miRNAs exhibited distinct rates of decrease (*Figure 4*). The results with elbow plot and k-means clustering analyses suggested that expression changes of miRNAs in this cluster could be divided into two groups, with one group consisting of miR-3 and miR-309 and the other containing miR-4, –5, −6 and −286 (*Figure 6—figure supplement 1*). We calculated the half-lives of these miRNAs and miR-3/–309 showed ~3–10 times faster degradation rates compared to those of miR-4, –5, −6 and −286 (*Figure 6B*). The faster disappearance of miR-3 and miR-309 was confirmed by Northern blotting analysis, excluding the possibility of sequencing artifacts (*Figure 6C and D*). It was possible that the unstable miRNAs were not loaded to the Argonaute complex hence more susceptible to degradation by ribonucleases. Therefore, we examined whether the miRNAs were properly loaded to the major miRNA Argonaute AGO1 (*Okamura et al., 2004*), by precipitating the AGO1 complex from two time windows (*Figure 6E*, *Figure 6—figure supplement 2*). We observed no obvious difference in the loading efficiency between stable (miR-4, –5, −6 and −286) and unstable (miR-3 and −309) miRNAs from this cluster. Furthermore, clear reduction of the mature miRNA species in the supernatant samples suggested that a large fraction of all the examined mature miRNAs were properly loaded to AGO1 (*Figure 6E* charts). Therefore, we concluded the difference in the reduction rate reflected the difference in the mature miRNA stability in the Argonaute complex.

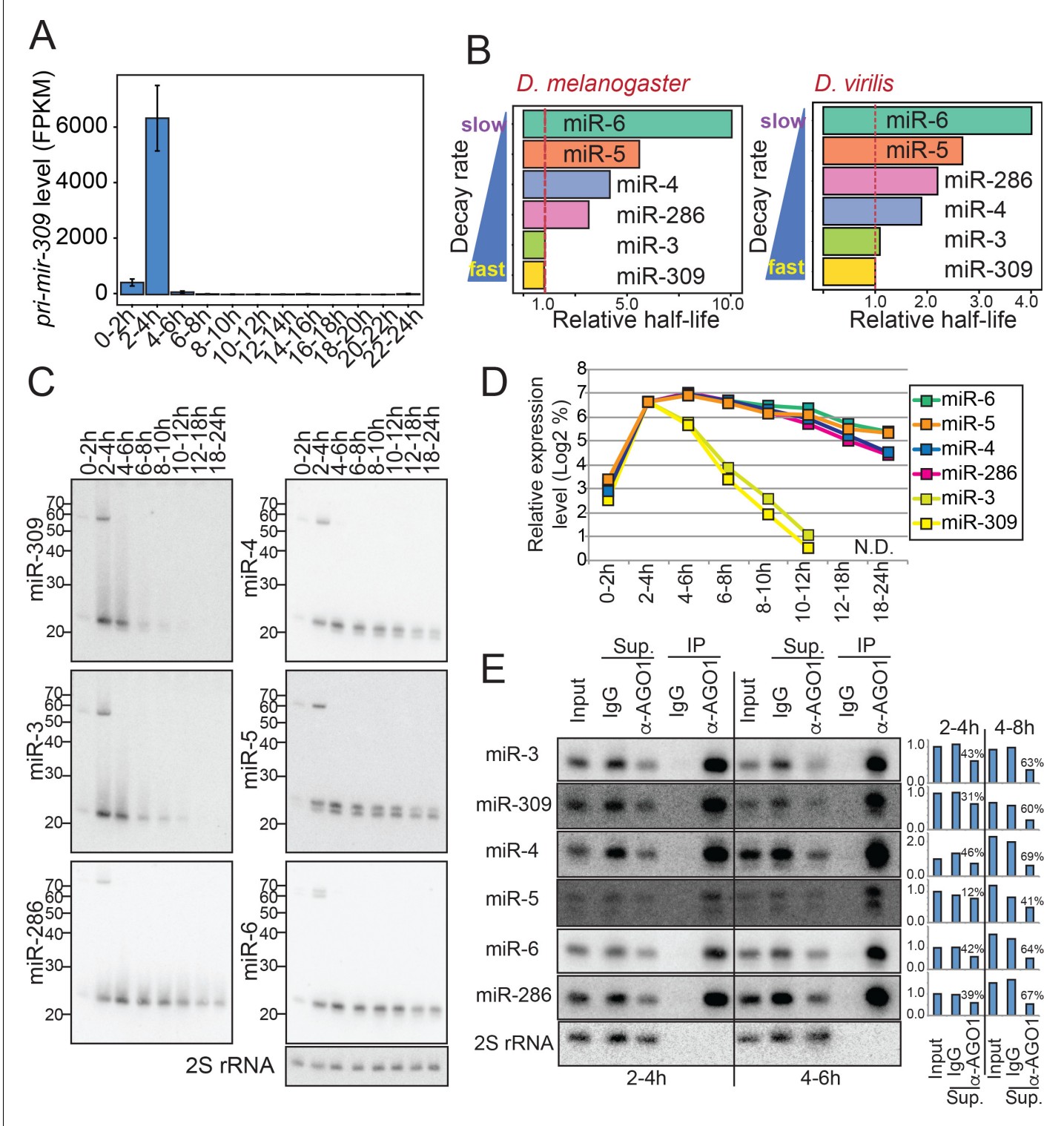

**Figure 6.** Differential mature miRNA half-lives for the *mir-309* cluster genes. (**A**) Levels of *pri-mir-309* during embryogenesis. Transcript levels were quantified using the total RNA-seq libraries and normalized by the FPKM (Fragment Per Kilobase of transcripts per Million mapped reads) method. *Pri-mir-309* is transiently expressed in 2–4 hr embryos. Error bars indicate the 95% confidence intervals. (**B**) Evolutionary conservation of the differential miRNA half-lives of *mir-309* cluster miRNAs. Estimated relative half-lives in *D. melanogaster* (this study) and *D. virilis* based on a published dataset (*Ninova et al., 2014*) are shown. The values are normalized by the half-life of miR-309. Note that *D. virilis* libraries did not include the spike-in oligos, and the TMM (Trimmed Mean of M-values) normalization was used (*Robinson and Oshlack, 2010*). The *D. melanogaster* data were normalized by the spike-in counts. (**C**) Northern blotting analysis was performed with the RNA samples extracted from embryos in the indicated time windows and using

*Figure 6 continued on next page*

*Figure 6 continued*

the probes detecting the indicated mature miRNA species. (**D**) Quantification of panel (**C**). The expression value of each miRNA in 0–2 hr sample was set as 100% and relative levels in each time window was calculated. The Y-axis shows log2 of % expression values. (**E**) Mature miRNA species from the *mir-309* cluster are efficiently loaded in the AGO1 complex. Lysates were prepared from 2-4 hr and 4–6 hr old embryos, and the AGO1 complex was precipitated using anti-AGO1 antibody. Rabbit IgG was used as a negative control. Efficient precipitation was confirmed by the enrichment of mature miRNA species in the AGO1-IP lane, and the depletion of the mature miRNA species in the AGO1-IP supernatant (Sup.) lane. The mature miRNA signals were quantified and normalized by the corresponding 2S rRNA signals in input and supernatant lanes. The input signal intensity at the 2–4 hr time window was used for further normalization for each miRNA species. Normalized values were plotted in the bar charts. The percentages in the charts indicate the degrees of mature miRNA depletion in the supernatant after AGO1-IP compared to the IgG control supernatant. We did not observe a correlation between miRNA stability and the degree of depletion by AGO1-IP, excluding the possibility that the differential half-lives of mature miRNAs from this cluster was caused by differential loading efficiencies of the miRNAs.

DOI: https://doi.org/10.7554/eLife.38389.019

The following figure supplements are available for figure 6:

**Figure supplement 1.** Elbow plot analysis for *mir-309* cluster.
DOI: https://doi.org/10.7554/eLife.38389.020

**Figure supplement 2.** Detection of AGO1 protein in immuno-precipitates used for *Figure 5 (E)*.
DOI: https://doi.org/10.7554/eLife.38389.021

## Biological importance of miR-3/–309 family miRNAs

We were interested in asking whether quick degradation of mature miR-3/–309 species is biologically important. Supporting the biological importance of distinct degradation rates, reanalysis of published small RNA library data from staged *D. virilis* embryos revealed a similar trend in the relative half-lives of the orthologous miRNAs (*Ninova et al., 2014*), suggesting that rapid down-regulation of miR-3/–309 miRNAs has evolutionarily conserved roles (*Figure 6B*).

miR-3 and miR-309 share the same seed sequence and are predicted to regulate largely overlapping sets of target mRNAs (*Figure 7A*)(*Bartel, 2009*). A plausible explanation is that down-regulation of some miR-3/–309 target mRNAs in early embryos is beneficial whereas high activity of miR-3/–309 in late embryos is detrimental to fly embryogenesis. Interestingly, a genome-wide overexpression screen showed that overexpression of miR-3 caused embryonic/early larval lethality, whereas none of the other members of this cluster caused early lethality when miRNAs in this cluster were individually overexpressed in embryos by the ubiquitous daughterless-Gal4 driver (*Bejarano et al., 2012*).

To gain further insight, we looked closely at the phenotypes of late embryos, and noticed that embryos overexpressing miR-3/–309 exhibited defects in denticle organization (*Figure 7B*). This phenotype is seen when the planar cell polarity (PCP) is affected, suggesting that overexpression of miR-3/–309 could cause PCP defects (*Donoughe and DiNardo, 2011*). To verify the PCP defects in a more established setting, we overexpressed miR-3 in the developing notum, where the polarity of sensory bristles serves as a faithful readout of PCP (*Lawrence et al., 2007*). Indeed, misexpression of miR-3 resulted in misorientation of bristles in the adult notum, demonstrating its biological activity in PCP in vivo (*Figure 7C*).

To investigate the underlying molecular mechanisms, we looked for potential targets using TargetScan (*Ruby et al., 2007*). Among predicted candidates that were expressed in embryos (*Westholm et al., 2014*; *Duff et al., 2015*), the target sites residing in the Van Gogh (Vang) 3' UTR caught our attention. Vang encodes a transmembrane protein that is integral to the Frizzled-mediated PCP pathway in various epithelial tissues (*Wolff and Rubin, 1998*; *Taylor et al., 1998*; *Marcinkevicius and Zallen, 2013*). To test whether Vang mRNA is a target of miR-3/–309 family miRNAs, we constructed a luciferase sensor by fusing the 3'UTR sequence of Vang downstream of the luciferase gene (*Figure 7D*). Expression of this sensor could be significantly repressed by co-overexpression of miR-3 (*Figure 7D*; wild-type sensor). This repression was dependent on the predicted target sites, because mutations at the predicted target sites abolished the repression (*Figure 7D*; mutant sensor). These results verified that Vang 3'UTR is a bona fide target of miR-3.

The hypothesis that rapid degradation of miR-3/–309 plays important biological roles implies the existence of miR-3/–309 target mRNAs that satisfy the following criteria: (1) up-regulated in the *mir-309* cluster mutant embryos only in early stages, (2) down-regulated in late embryos overexpressing miR-3 or miR-309. Having established the miRNA-target relationship between miR-3 and Vang, we

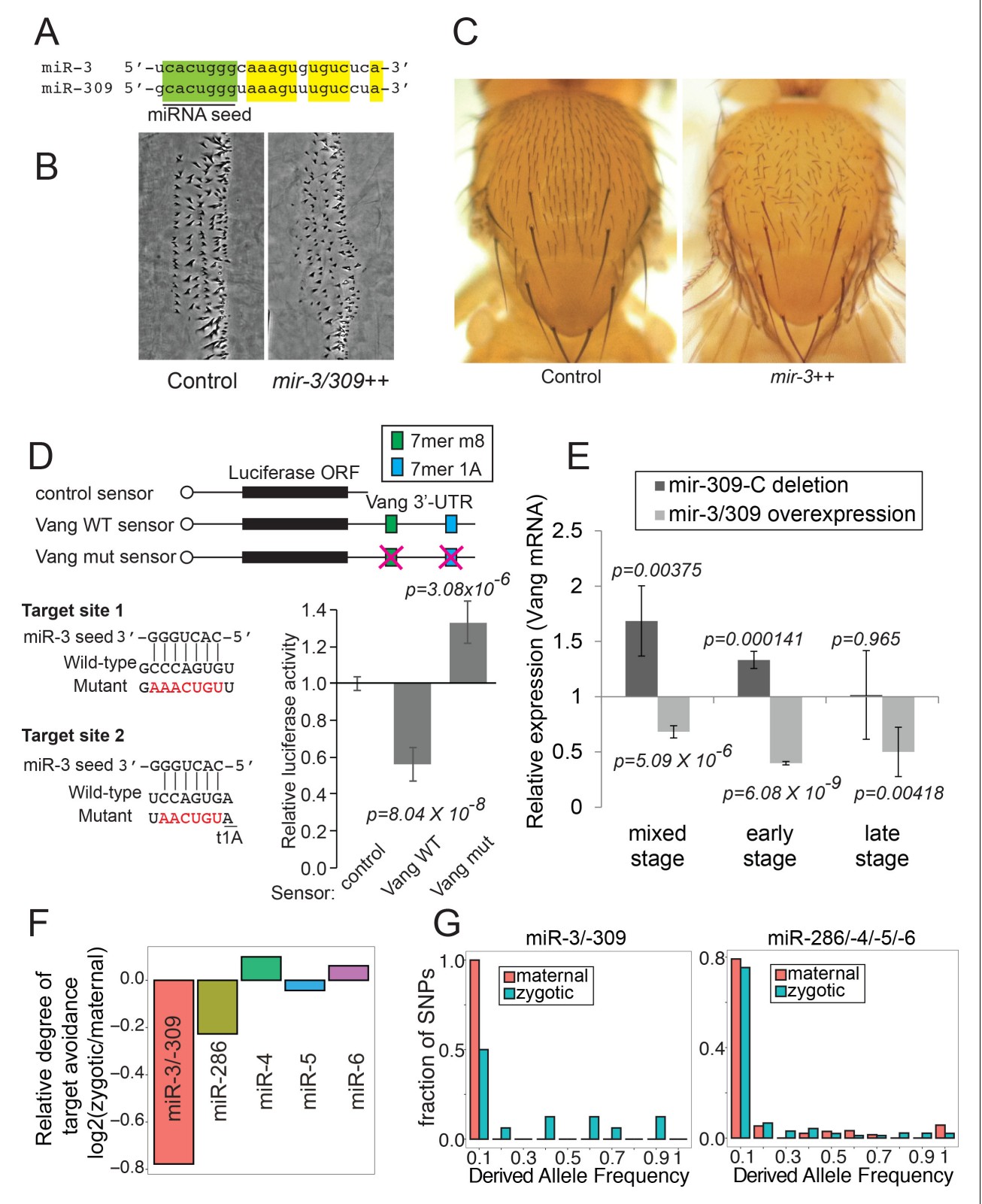

**Figure 7.** Biological activity of miR-3/–309. (**A**) Sequences of mature miR-3 and miR-309. Note that their seed sequences are identical. (**B**) Embryonic denticles in control and embryos overexpressing miR-3/–309 (da-gal4 - > UAS-mir-3/–309). Misorientation of denticles was observed in embryos overexpressing miR-3/–309. (**C**) Misorientation of adult sensory bristles on the adult notum by miR-3 overexpression (eq-gal4 - > UAS-mir-3). (**D**) Luciferase sensor assays. S2-R+ cells were transfected with plasmids carrying a luciferase sensor containing the Vang 3' UTR sequence along with a

*Figure 7 continued on next page*

*Figure 7 continued*

plasmid to overexpress miR-3 or a negative control empty vector. The averages and standard deviations of normalized Luciferase activity are shown (N = 8). The p-values were calculated by t-test. Vang 3' UTR contains two predicted miR-3 family target sites. Removal of these predicted target sites abolished down-regulation of the Vang sensor by overexpressed miR-3. Raw data can be found in *Figure 7D*-source data 1. (E) Expression levels of Vang mRNA determined by quantitative RT-PCR. Values were normalized against RpL32. Embryos were collected at three time windows (mixed: 0–24 hr, early:0–12 hr or late:12–24 hr) using control, a deletion mutant lacking the *mir-309* cluster (*mir-309-C*) miRNAs (dark gray) and a transgenic line overexpressing miR-3/–309 (light gray). The average relative expression ±standard deviation and p-values are shown (t-test). Vang mRNA is up-regulated in the deletion mutant whereas down-regulated in embryos overexpressing miR-3/–309 in the mixed stage samples. Note that derepression of Vang was not significant in null mutant embryos in the late stage consistent with the low level of miR-3/–309 in late embryos due to quick degradation. (F) To calculate relative target avoidance, the fractions of polymorphic target sites with target allele frequencies < 0.1 were computed for maternal and zygotic mRNAs. The ratio of fractions (Fraction$_{zygotic}$/Fraction$_{maternal}$) was defined as relative degree of target avoidance. miR-3/–309 polymorphic target sites found in zygotically expressed mRNAs showed a weaker degree of target avoidance compared to those in maternal mRNAs, while much smaller differences were seen in the degree of target avoidance for target sites of other members of *mir-309* cluster miRNAs. (G) Distributions of derived target allele frequencies (DAF) on maternal and zygotic genes for miR-3/–309 targets (left) or miR-286/–4/−5/–6 (right) were plotted. The zygotic DAF distribution for miR-3/–309 targets was shifted to the right compared to that of maternal genes (D = 0.56, p=0.046, N$_{maternal}$ = 7, N$_{zygotic}$ = 16; one-tailed Kolmogorov-Smirnov test). The difference was less significant for the miR-286/–4/−5/–6 target set (D = 0.18, p=0.078, N$_{maternal}$ = 52, N$_{zygotic}$ = 131). Raw data for (F) and (G) are reported in *Supplementary file 6*.
DOI: https://doi.org/10.7554/eLife.38389.022

The following source data and figure supplements are available for figure 7:

**Source data 1.** Normalized Renilla (sensor)/firefly (control) luciferase activity ratios that were normalized to the psiCHECK empty vector value for each of the pDsRed and pDsRed-miR-3 groups are shown in the first 13 rows.
DOI: https://doi.org/10.7554/eLife.38389.026
**Figure supplement 1.** Levels of miR-3/–309 overexpression.
DOI: https://doi.org/10.7554/eLife.38389.023
**Figure supplement 2.** Expression levels of miR-3, miR-309 and Vang mRNA.
DOI: https://doi.org/10.7554/eLife.38389.024
**Figure supplement 3.** Allele frequency distribution of polymorphic miRNA seed target sites.
DOI: https://doi.org/10.7554/eLife.38389.025

were interested in testing whether Vang satisfies these conditions. To this end, we quantified the levels of Vang mRNA in wild-type and the *mir-309* cluster null mutant (*Bushati et al., 2008*) embryos, as well as in embryos overexpressing miR-3/–309 (*Suh et al., 2015*) (*Figure 7E*). Consistent with the sensor results (*Figure 7D*), Vang mRNA was up-regulated in the null mutant and down-regulated upon overexpression of miR-3/–309 when mixed stage embryos were used (*Figure 7E* left). As predicted, Vang mRNA expression was not significantly altered in mutant embryos in late stages where miR-3 and miR-309 were undetectable due to quick degradation (*Figures 5C* and *7E* right). Furthermore, Vang mRNA expression could be repressed in late embryos when miR-3/–309 was overexpressed (*Figure 7E* right).

We further verified that the levels of overexpressed miR-3/–309 products did not exceed the natural levels of these miRNAs at the expression peaks (*Figure 7—figure supplement 1*), confirming that the level of overexpression was within the physiological range. Reanalysis of RNA-seq data indicated that the reduction of miR-3/–309 preceded the increase of Vang mRNA, consistent with our hypothesis (*Figure 7—figure supplement 2*). Taken together, these observations support the notion that rapid degradation of miR-3/–309 is biologically important.

## Effects of miRNA stability on target site evolution

Due to the regulatory activity of miRNAs against 3'UTRs harboring seed target sites, 3'UTRs tend to avoid seed target sequences of miRNAs that are expressed in the same tissue (*Stark et al., 2005*; *Farh et al., 2005*). In turn, this target avoidance leaves evolutionary signatures that could be detected by population genetics analysis (*Chen and Rajewsky, 2006*; *Marco, 2015*).

Since miR-3 and miR-309 share the same seed sequence and are present at very low levels in late embryos, we hypothesized that miR-3/–309 target sites would be less strongly selected against in 3'UTRs of zygotically expressed genes compared to those of the other cluster members. We analyzed the *D. melanogaster* SNP (single nucleotide polymorphism) data generated from 205 inbred lines (DGRP: *D. melanogaster* Genetic Reference Panel) (*Mackay et al., 2012*; *Huang et al., 2014*). We tested whether miRNA target sites on 3'UTRs of maternally and zygotically expressed genes

show distinct behaviors. Maternally and zygotically expressed mRNAs were defined in a previous study (*Paris et al., 2015*). We first plotted distributions of allele frequencies of polymorphic miRNA target sites for each miRNA seed sequence (*Figure 7—figure supplement 3*). Due to random selection in the fly population, two alleles with neutral effects are predicted to show a symmetric 'U-shaped' distribution of allele frequencies with most individuals homozygous for each allele (*Marco, 2015*). Selective pressure for or against an allele would shift the U-shaped distribution to the right or left, respectively.

To estimate the relative selective pressure against miRNA target sites in 3'UTRs between maternal and zygotic genes, the ratios of fractions of the target alleles in the 0–0.1 bin between zygotic and maternal mRNAs were calculated (*Figure 7F*, *Figure 7—figure supplement 3* and *Supplementary file 6*). The ratio between zygotic/maternal target allele fractions showed lower values with miR-3/–309 target sites compared to the other four seed sequences, consistent with our hypothesis that zygotic genes would avoid miR-3/–309 target sites less strongly compared to seed target sites of the other miRNAs from this cluster. To verify that this is a result of the selective pressure against target sites, we turned to DAF (Derived Allele Frequency) analysis. We first identified derived miRNA target sites that are non-target sites in the ancestral state, and plotted the allele frequency. We found that the DAF was generally lower for maternal mRNAs than zygotic mRNAs for miR-3/–309 target sites, whereas target sites for the other four miRNAs in maternal and zygotic gene sets showed more similar DAF distributions (*Figure 7G*, *Supplementary file 6*).

These results demonstrate that miRNA stability has detectable effects on target site evolution in the transcriptome. Together with the experimental evidence suggesting the significance of rapid degradation of miR-3 and miR-309 in target regulation in embryos (*Figure 7B–C*), our results highlight the importance of miRNA degradation rates in shaping gene regulatory networks and their evolution.

## Discussion

### Dynamic changes of the total miRNA abundance during fly embryogenesis

Spike-in normalization of small RNA library data allowed us to understand changes in the total abundance of small RNA families during embryogenesis (*Figure 1B*). As expected, piRNA reads gradually decreased during embryogenesis, consistent with previous reports (*Brennecke et al., 2008*). This was associated with concomitant decrease of the mRNA levels of piwi-clade Argonaute genes (*Figure 1—figure supplement 2*). On the other hand, the total abundance of miRNAs kept increasing (*Figure 1B*). This trend with lower miRNA abundance in the fertilized eggs and continuous increase during embryogenesis appears to be conserved broadly including mammals (*Ohnishi et al., 2010*). However, mRNA levels of the miRNA processing factors were relatively low in late embryos (*Figure 1—figure supplement 2*). It is unclear how embryos can support the dramatic (~8-fold) increase of total miRNA level. One possibility is that general miRNA processing activity is not a rate-limiting factor, and transcription activity of individual miRNA genes largely determines the levels of mature miRNAs. This is consistent with our results that miRNA expression changes could be predicted by the level of transcription and the level of mature miRNA at a given moment, which suggested that miRNA production rate per pri-miRNA molecule and degradation rate generally stayed constant during fly embryogenesis (*Figure 1D,E*).

### Expression of miRNAs from polycistronic genes

miRNAs often form gene clusters and miRNA hairpins within a cluster are generally co-transcribed. Although it is unclear why miRNA genes are often clustered, recent biochemical studies provide some clues. Some miRNA cluster transcripts form intramolecular structures that interfere with processing of miRNA hairpins by Drosha, and in some cases, additional processing steps are required to resolve the structures (*Du et al., 2015*; *Chaulk et al., 2014b*; *Chaulk et al., 2014a*; *Chaulk et al., 2011*). In addition, a recent study suggests that processing efficiency of clustered miRNAs show a positive correlation with the distance from TSS, potentially due to interaction between the microprocessor complex and the transcription machinery in a manner dependent on phosphorylation status

of the Pol II-CTD (C-Terminal Domain) (*Church et al., 2017*). Therefore, the clustered configuration of miRNA genes likely adds another layer of regulation to miRNA processing.

Due to the lack of sensitive in situ detection methods of mature miRNA species in fly embryos and tissues, spatial expression patterns were mainly studied by in situ hybridization against primary miRNAs or detecting reporter genes fused with the regulatory elements of miRNA genes (*Aboobaker et al., 2005*; *Brennecke et al., 2003*). Our study suggested that the transcriptional activity of the miRNA locus is an important determinant of miRNA gene expression (*Figure 1*), supporting the validity of such pri-miRNA analyses for understanding miRNA expression. However, results of these approaches need to be carefully interpreted considering several factors: (1) alternative pri-miRNA isoforms: miRNA genes can utilize multiple TSSs and TTSs to generate alternative pri-miRNA isoforms (*Chang et al., 2015*; *de Rie et al., 2017*). Depending on the probe design for pri-miRNA detection, only subsets of pri-miRNA isoforms may be detectable. (2) miRNA stability: pri-miRNAs are often transiently expressed (*Figure 1C*) and mature miRNA stability has significant effects on the mature miRNA steady state level.

The *mir-317* cluster provides an interesting example. In *Drosophila*, expression of *mir-34* is induced in aged flies compared to young adults, and *mir-34* null mutant exhibits shortened lifespan with evidence for progressive neurodegeneration in the central nervous system (*Liu et al., 2012*). The function of miR-34 in regulating longevity is at least in part mediated by down-regulation of Eip74EF, a transcription factor that mediates down-regulation of ecdysone-repressed genes (*Shlyueva et al., 2014*). Therefore, a plausible possibility is that *mir-34* and the ecdysone pathway form a double negative feedback loop to create a bistable switch by using the promoter at the upstream TSS of the *mir-317* cluster as speculated previously (*Liu et al., 2012*; *Sempere et al., 2002*; *Xiong et al., 2016*). This hypothesis is supported by our observation that ecdysone treatment leads to a decrease in the miR-34 level, favoring the use of TSS3 for the transcription and maturation of miR-277 alone (*Figure 5B,C*, *Figure 5—figure supplement 3*). Besides its well-known roles in metamorphosis, the ecdysone pathway plays a role in regulation of longevity (*Simon et al., 2003*), and the miR-34/Eip74EF axis may be an upstream regulatory mechanism of this pathway.

Another cluster member miR-277 appears to play roles in longevity in a different way. Alteration of miR-277 activity resulted in short lifespans by indirectly modulating the activity of TOR (target of Rapamycin) signaling (*Esslinger et al., 2013*). Notably, the level of miR-277 expression was reduced in aged flies compared to young adults, in contrast to the level of miR-34. The independent TSS for *mir-277* specific pri-miRNA isoform could enable the independent regulation of miRNAs in a single cluster. The biological significance of the gene organization remains unclear, however, the organization of the homologous *mir-317* cluster is conserved in *Capitella*, which is estimated to have split from the Ecdysozoan branch >600 million years ago (*Peterson et al., 2009*). This may suggest that the gene arrangement is essential for proper regulation of individual genes in the cluster. Similarly, alternative selection of TSS in miRNA clusters has been noted in mammalian systems for the *let-7* cluster miRNAs, which is another deeply conserved miRNA cluster whose gene order is also generally conserved (*Chang et al., 2015*). Complex cis-regulatory elements embedded in miRNA clusters may impose evolutionary constraints that maintain the gene order, and may explain why the miRNA cluster configuration is often preserved during evolution.

## Mature miRNA degradation and its importance in gene regulation

The *mir-309* cluster is known to be a fast-evolving miRNA cluster with frequent gene duplications and rearrangements leading to gains or losses of individual cluster member homologs during insect evolution (*Ninova et al., 2014*; *Mohammed et al., 2013*). Homologs of the *mir-309* cluster members show similar expression patterns in Drosophilids, but in mosquitoes, the gene organization and apparent functions are different. In *Aedes aegypti*, expression of miR-309 is transiently upregulated in the ovary after blood feeding that triggers oocyte maturation in mosquitoes (*Zhang et al., 2016*). The level of miR-309 peaks at 36 hr post blood-meal (hPBM) and sharply decreases by 48hPBM. This suggests that the short half-life of miR-309 is conserved in other dipteran insects and in different tissues, even though the biological functions have diverged. Its short half-life may be suitable for gene regulation during fast processes like oogenesis and early embryogenesis.

The evolutionary patterns of miR-3/–309 target sites and misregulation of Vang in late embryos upon miR-3 overexpression suggest the importance of rapid degradation of these two miRNAs (*Figure 7*). There are known sequence-specific miRNA destabilization phenomena although the

underlying mechanisms are not completely understood (*Rüegger and Großhans, 2012*). Future studies should aim to elucidate molecular mechanisms of miR-3/–309 destabilization and identification of such factors will allow us to experimentally study the effects of miR-3/–309 destabilization.

# Materials and methods

**Key resources table**

| Reagent type (species) or resource | Designation | Source or reference | Identifiers | Additional information |
|---|---|---|---|---|
| Genetic reagent (*D. melanogaster*) | Eq-Gal4 | Bloomington Drosophila Stock Center | BDSC:43659 | FlyBase symbol: P{GAL4-Hsp70.PB}l(3)Eq1Eq1 |
| Genetic reagent (*D. melanogaster*) | Da-Gal4 | Bloomington Drosophila Stock Center | BDSC:55851 | FlyBase symbol: P{GAL4-da.G32}UH1 |
| Genetic reagent (*D. melanogaster*) | UAS-mir-3 | PMID: 22745315 | | |
| Genetic reagent (*D. melanogaster*) | UAS-mir-3/–309 | PMID: 26138755 | | |
| Genetic reagent (*D. melanogaster*) | mir-309-C delta 1 | PMID: 18394895 | | |
| Cell line (*D. melanogaster*) | S2-R+ | PMID: 9822716 | DGRC Cat# 150; RRID:CVCL_Z831 | S2-R + cells maintained in the Lai and Okamura labs |
| Cell line (*D. melanogaster*) | Kc167 | Drosophila Genomics Resource Center | DGRC Cat# 1; RRID:CVCL_Z834 | |
| Cell line (*D. melanogaster*) | BG3-c2 | Drosophila Genomics Resource Center | DGRC Cat# 6;, RRID:CVCL_Z728 | |
| Antibody | Anti-AGO1 (rabbit polyclonal) | Abcam | Abcam Cat# ab5070; RRID:AB_2277644 | 1:1000 in TBST |
| Antibody | Anti-alpha-tubulin (mouse monoclonal clone DM1A) | Sigma | Sigma-Aldrich Cat# T9026; RRID:AB_477593 | 1:1000 in TBST |
| Recombinant DNA reagent | psiCHECK2 (modified MCS) | PMID:17599402 | | SalI-SacI-NotI-XbaI-SalI-EcoRI-EcoRV-XhoI-SpeI sites were inserted in the SalI/NotI sites of psiCHECK2 (Promega) |
| Recombinant DNA reagent | psiCHECK-Vang wild-type | This study | | Vang 3'UTR was amplified from genomc DNA using NotI_Vang_2128 and XhoI _Vang_3492 primers |
| Recombinant DNA reagent | psiCHECK-Vang mutant | This study | | The two mir-3 target sites were mutated by overlap PCR using Vang_t1_mut_F, Vang_t1_mut_R, Vang_t2_mut_F and Vang_t2_mut_R and the cloning primers used for cloning of the wild-type sensor |
| Recombinant DNA reagent | pUAST-DsRed-mir-3 | PMID: 22745315 | | |
| Recombinant DNA reagent | pUAST-DsRed | PMID: 12679032 | | |
| Recombinant DNA reagent | pRmHa-mir-283-C | This study | | Genomic DNA fragment amplified by EcoRI_dme_mir283c_F and SalI_dme_mir283c_R inserted to pRmHa3 (DGRC: 1145). |
| Recombinant DNA reagent | pRmHa-mir-283 A-to-U | This study | | Site directed mutagenesis of pRmHa-mir-283-C using dme_mir283_5pAtoU_F and dme_mir283_5pAtoU_R |

*Continued on next page*

*Continued*

| Reagent type (species) or resource | Designation | Source or reference | Identifiers | Additional information |
|---|---|---|---|---|
| Recombinant DNA reagent | pRmHa-mir-283 A-to-G | This study | | Site directed mutagenesis of pRmHa-mir-283-C using dme_mir283_5pAtoG_F and dme_mir283_5pAtoG_R |
| Recombinant DNA reagent | pRmHa-mir-283 A-to-C | This study | | Site directed mutagenesis of pRmHa-mir-283-C using dme_mir283_5pAtoC_F and dme_mir283_5pAtoC_R |
| Recombinant DNA reagent | pRmHa-mir-92b | This study | | Genomic DNA fragment amplified by mir92b_genespecificF and mir92b_genespecificR inserted to pRmHa3 (DGRC: 1145) |
| Recombinant DNA reagent | pRmHa-mir-92b A-to-U | This study | | Site directed mutagenesis of pRmHa-mir-92b using dme_mir92b_5pAtoU_F and dme_mir92b_5pAtoU_R |
| Recombinant DNA reagent | pRmHa-mir-92b A-to-G | This study | | Site directed mutagenesis of pRmHa-mir-92b using dme_mir92b_5pAtoG_F and dme_mir92b_5pAtoG_R |
| Recombinant DNA reagent | pRmHa-mir-92b A-to-C | This study | | Site directed mutagenesis of pRmHa-mir-92b using dme_mir92b_5pAtoC_F and dme_mir92b_5pAtoC_R |
| Recombinant DNA reagent | pRmHa-mir-263a | This study | | Genomic DNA fragment amplified by mir263a_genespecifiF and mir263a_genespecifiR inserted to pRmHa3 (DGRC: 1145) |
| Recombinant DNA reagent | pRmHa-mir-263a A-to-U | This study | | Site directed mutagenesis of pRmHa-mir-263a using dme_mir263a_5pAtoU_F and dme_mir263a_5pAtoU_R |
| Recombinant DNA reagent | pRmHa-mir-263a A-to-C | This study | | Site directed mutagenesis of pRmHa-mir-263a using dme_mir263a_5pAtoC_F and dme_mir263a_5pAtoC_R |
| Commercial assay or kit | Dual-Glo Luciferase Assay System | Promega | Promega:E2940 | |
| Commercial assay or kit | Effectene | QIAGEN | QIAGEN: 301427 | |
| Chemical compound, drug | 20-Hydroxyecdysone | Sigma | Sigma: H5142 | |
| Chemical compound, drug | Bathocuproinedisulfonic acid disodium salt | Sigma | Sigma: B1125 | |
| Software, algorithm | FASTX-toolkit | Hannon Lab | | http://hannonlab.cshl.edu/fastx_toolkit |
| Software, algorithm | Bowtie1.1.2 | PMID: 19261174 | | |
| Software, algorithm | STAR | PMID: 23104886 | | |
| Software, algorithm | Cufflinks suite tools | PMID: 20436464 | | |
| Software, algorithm | UCSC liftOver | UCSC Genome Browser | | https://genome.ucsc.edu/cgi-bin/hgLiftOver |

## Small RNA library construction and bioinformatics analysis

For small RNA library construction, we prepared RNA samples from staged embryos in biological triplicate. Two sets of libraries were prepared and sequenced together, while the other replicate

was prepared independently from the first two sets. Small RNA libraries were prepared as previously described (*Lim et al., 2016*), and sequenced at Duke-NUS Genomics facility or BGI on Hiseq2000 or Hiseq4000. The spike-in sequences are shown in *Supplementary file 7*. After the read quality was checked by FastQC (http://www.bioinformatics.babraham.ac.uk/projects/fastqc), the adaptor sequence was trimmed off by FASTX-toolkit (http://hannonlab.cshl.edu/fastx_toolkit) and 18-30nt reads were mapped to dm3 genome using Bowtie 1.1.2 with no mismatch allowed (*Langmead et al., 2009*). Reads corresponding to 2S rRNA were removed prior to genome mapping. Reads corresponding to four categories (abundant ncRNA, miRNA, siRNA/piRNA and other genome mapping reads) were identified sequentially by mapping reads to the reference sequences without double counting as described previously (*Chak et al., 2015*). Spike-in reads and reads mapping to miRNA arms were identified and counted by mapping small RNA reads to the spike-in sequences (*Supplementary file 7*) and miRNA sequences defined in (*Lim et al., 2016*), respectively. Spike-in read counts were used as a normalizer and normalized values were expressed in RPTS (Reads per thousand spike-in reads). To quantify TE-siRNAs and TE-piRNAs, we defined 21nt reads and 23-30nt reads as siRNAs and piRNAs, respectively. *D. virilis* miRNA read counts were previously published and values were normalized by the TMM (Trimmed Mean of M-values) method (*Robinson and Oshlack, 2010*; *Ninova et al., 2014*). miRNA half-lives were estimated by fitting the normalized read count values to the log-linear model. To detect time windows with distinct expression change rates between cluster members, we first calculated the change rates in every two consecutive time windows followed by ANOVA test. The P values for multiple comparisons were further adjusted by the Benjamini-Hochberg method to control the false discovery rate (*Benjamini and Hochberg, 1995*). For the charts in *Figure 3*, miRNA genes were removed if (1) miRNAs with average RPTS from all time windows is <5 or (2) the origin of mature miRNA reads could not be determined due to gene duplication. For miRNA paralogs from the same cluster like *mir-6–1/−2/–3*, the average value was used. According to these cutoffs, the following miRNAs were excluded from the analyses: *mir-281–1/−281–2* cluster, *mir-13a/–13b-1/–2* c cluster, *mir-2a-1/−2a-2/−2b-2* cluster, *mir-2499/–4966/−972 ~ 979* cluster, *mir-959 ~964* cluster, *let-7/mir-100/–125* cluster, *mir-318/–994* cluster, *mir-303/–982 ~ 984* cluster, *mir-992/–991/−2498* from *mir-310* cluster.

## Computational analysis of total RNA-seq libraries

Published modENCODE total RNA-seq libraries (*Brown et al., 2014*; *Westholm et al., 2014*; *Duff et al., 2015*) were downloaded from NCBI. The accession numbers are listed in *Supplementary file 1*. We first removed reads matching to the rDNA sequence (NCBI accession number M21017.1) and mapped remaining reads to the dm3 genome using STAR alignment tool (*Dobin et al., 2013*). We used the following setting for genome mapping: '–outFilterMismatchNoverLmax 0.02 –alignIntronMin 20 –alignIntronMax 200000'. We used Flybase 6.06 gene annotation for the definition of pri-miRNA sequences (*Attrill et al., 2016*). For quantification of pri-miRNA isoforms, we constructed a mini-genome containing 50 kb each up- and downstream genomic sequences of the miRNA hairpin and mapping reads were quantified by cufflinks with the following setting: '–library-type fr-firststrand –min-intron-length 20 –max-intron-length 200000' and other cufflinks suite tools (*Trapnell et al., 2010*). When unannotated pri-miRNAs spanning miRNA hairpins were present in the miRNA genomic loci, we manually modified gene models or added new isoforms based on the transcripts in the merged mapping data using all embryo libraries with cuffmerge from the tuxedo suite (*Trapnell et al., 2010*). Manually added pri-miRNA isoforms are listed in *Supplementary file 8*.

## Multiple regression analysis

For each data point of each miRNA gene, three values were calculated from two consecutive time windows using the following definitions. $x$ is the initial miRNA level, which is the mature miRNA level for the first of the two time windows. $z$ is the miRNA change rate, which is the difference in the mature miRNA level between the two windows (second window – first window). $y$ is the upstream density, which is the pri-miRNA read density in the 100nt region immediately upstream of the miRNA hairpin in the second window of the total RNA-seq libraries. For calculation of the pri-miRNA read density in 12–18 hr and 18–24 hr windows, we used the averages of total RNA-seq data from the three 2 hr time windows. Shown are genes that satisfy the following criteria: normalized read

abundance in 0-2hAEL constitute >30% of the sum of RPTS from all windows, upstream read density >1 in>=1 time windows and mature miRNA >10 RPTS in >= 5 time windows. We also removed miRNA genes when > 20% reads mapped to multiple genomic positions, to avoid artifacts due to incorrect assignment of the gene origin. Coefficients a, b and c were obtained by fitting the x, y and z values of mature and primary miRNAs to the following equation: $z \sim ax + by + c$ For the analysis shown in *Figure 3—figure supplement 1*, the coefficient values associated with mature miRNA level (Column 'Coef_mature_Level' in *Supplementary file 4*) were used.

## miRNA target and population genetics analyses

The protocol is similar to that used in *Marco (2015)* . In brief, the single-nucleotide polymorphisms (SNPs) data from Drosophila Genetic Reference Panel (*Mackay et al., 2012*; *Huang et al., 2014*) were mapped against 3' UTR sequences of *Drosophila melanogaster* release 5.57 from FlyBase. 18 non-target variants for each miRNA (hexamer sequences that are different from the miRNA seed target sequence by one nucleotide) were computationally constructed according to the previously established method (*Marco, 2015*). SNPs that are from any of the 18 non-target variants for each miRNA were kept. SNPs that are overlapped with protein-coding sequences were discarded. For each polymorphic target site, the allele frequency was calculated as the proportion of the target allele within the sampled population ($N_{target\_allele} / (N_{non-target\_allele} + N_{target\_allele})$). Maternal gene set was defined by a previous study (*Paris et al., 2015*), and the remaining genes were used as zygotic genes. The expression levels of genes were quantified by cufflinks (*Trapnell et al., 2010*) using the total RNA-seq data (*Westholm et al., 2014*; *Duff et al., 2015*) and genes with maximum FPKM >0.1 were considered for further analyses. To study the derived allele frequencies (DAFs), the coordinates of polymorphic target sites from *D. melanogaster* genome were first lifted to *Drosophila sechellia* by UCSC liftOver (https://genome.ucsc.edu/cgi-bin/hgLiftOver). The polymorphic target sites whose counterparts in *D. sechellia* are non-targets were considered as derived target sites.

## RNA extraction, Northern blotting and immunoprecipitation

Canton S embryos were collected on grape juice plates supplemented with yeast paste and RNA was extracted at the indicated time window. Northern blotting was performed as previously described (*Okamura et al., 2007*). A pre-stained ladder (BioDynamics) or a radioactive ladder (Ambion) was run concurrently with the RNA samples on denaturing polyacrylamide gels.10 μg total RNA was loaded on each lane and membranes were hybridized with the probes listed in (*Supplementary file 7*). Immunoprecipitation was performed according to the protocol published previously using 5 μg anti-AGO1 antibody (AbCam) immobilized on 32 μl Dynabeads (Invitrogen) and 400 μl total lysate prepared from 2–4 hr or 4–6 hr old embryos using RIPA buffer (1x PBS pH 7.4, 0.1% SDS, 0.5% deoxycholate, 0.5% NP40) (*Okamura et al., 2007*). Extracted RNA was separated by denatured PAGE and probed with the indicated probes. Probe sequences or LNA probe product numbers are summarized in *Supplementary file 7*. For analysis of total RNA or AGO1 protein abundance per embryo, 100 or 50 embryos were collected and homogenized in 100 ul Trizol or 25 ul 2xSDS-PAGE buffer, respectively, without dechorionation.

## Plasmid construction, mutagenesis and transfection

The *mir-92b*, *mir-263a*, or *mir-283* genomic fragment was cloned into a plasmid driven by a copper sulphate inducible metallothionein promoter. 5'A-to-U, 5'A-to-5'C, or 5'A-to-5'G mutants were generated by PCR followed by DpnI digestion or by overlap PCR. S2-R+ cells were seeded at $1 \times 10^6$ cells/ml on T75 flasks and transfected with either 5.625 μg of wild type or 5' mutant plasmid using Effectene (Qiagen). Cells were washed thrice with fresh medium and plated onto new flasks 24 hr following transfection. 1.5 mM CuSO$_4$ was added to induce expression or 50 μM bathocuproinedisulfonic acid disodium salt (BCS) was added (for control). 24 hr after induction, cells were washed twice with fresh medium containing 500 μM BCS and re-suspended in medium containing 50 μM BCS. Cells were washed with 1 x PBS prior to RNA extraction using Trizol (Invitrogen) at indicated time points after BCS addition. Northern blotting was performed as described above. Oligos and probes used are summarized in *Supplementary file 7*.

## Cell culture and 20-HE stimulation

Cell culture was done according to the protocols established by the DGRC (Drosophila Genome Resource Center). S2-R+ cells were grown in Schneider's *Drosophila* medium (Invitrogen) supplemented with 10% heat-inactivated Fetal Bovine Serum (FBS) and 1% Penicillin-Streptomycin. Kc167 cells were grown in Shields and Sang M3 insect medium (Sigma) supplemented with 0.25% Bacto Peptone, 0.1% Yeast Extract, and 10% FBS. BG3-c2 cells were grown in Shields and Sang M3 medium (Sigma) supplemented with 10% FBS and 20 µg/ml insulin (Sigma). 20-hydroxyecdysone (20-HE) stimulation was done by adding 20-HE at the final concentration of 5 µM and incubating cells for the indicated times. The presence of mycoplasma in the cell lines used in this study (S2-R+, Kc167 and BG3-c2) was tested by Universal Mycoplasma Detection Kit (ATCC, #30-1012K) and no mycoplasma contamination was detected.

## Luciferase sensor assays

To construct the wild-type Vang sensor, we amplified the 3'UTR of Vang using the PCR primers (NotI_Vang_2128 and XhoI_Vang_3492, *Supplementary file 4*) and genomic DNA as a template, and the fragment was inserted to the Not I and Xho I sites of the modified psiCHECK2 plasmid (*Okamura et al., 2007*). The mutant sensor was constructed by overlapping PCR using the mutagenesis primers shown in *Supplementary file 4*. Luciferase assays were performed using S2-R+ cells according to the previously published protocol (*Okamura et al., 2007*). Briefly, S2-R+ cells were seeded at $1 \times 10^6$ cells/ml in a 96-well plate, and cells were transfected using Effectene (Qiagen). 25 ng of the sensor plasmid, 25 ng of the miRNA expression plasmid and 12.5 ng of the Ub-Gal4 plasimid was used for each well. Dual-Luciferase Reporter (DLR) assay (Promega, USA) was carried out to quantitate the effects of miR-3 overexpression on the respective Vang sensors. The assay was performed according to the manufacturer's protocol. Assays were performed in quadruplicate at each time, and repeated twice on different days. The two sets of data were combined to draw the charts.

## Overexpression of miR-3 in the notum and embryo

Overexpression of miRNAs in the developing adult notum was performed according to the previous study using eq-gal4 as a driver (*Bejarano et al., 2012*). For overexpression of miR-3/–309 in embryos, the UAS-mir-3/mir-309 line (*Suh et al., 2015*) was crossed with da-gal4 and aged embryos were used for RNA extraction and cuticle analysis. Total RNA was extracted for each experimental condition using the RNeasy Mini Kit (Qiagen) as per the manufacturer's protocol. Total RNA concentration was measured using NanoDrop ND-2000 Spectrophotometer and the purity of the samples was determined by the OD ratios, $A_{260}/A_{280}$. One µg of total RNA was reverse transcribed in a 20 µl reaction volume using the QuantiTect reverse transcription kit (Qiagen) according to the manufacturer's protocol. Quantitation of mRNA was performed using SYBR Green Assay (Thermo Fisher Scientific) on the PikoReal Real-Time PCR System (Thermo Fisher Scientific) and a PCR product dissociation curve was generated to ensure specificity of amplification. *RpL32* was used as an endogenous control and relative quantitation was performed using relative quantification ($2^{-\Delta\Delta CT}$). Results were generated from three technical replicates. The average relative expression ± standard deviation (SD) was determined and two-sample t-test was carried out to determine statistical significance. Primers used for qPCR reactions are summarized in *Supplementary file 7*.

## Accession number

The small RNA library data produced for this study are deposited at NCBI SRA under SRP109269.

## Acknowledgements

The authors are grateful to members of KO and GT-K laboratories for discussion. We thank Li-Ling Chak for her help with initial small RNA library analysis and Duke-NUS Genomics facility and BGI for Illumina sequencing. We thank DGRC (Drosophila Genome Resource Center) for cell lines, Dr. Walton Jones at KAIST for the UAS-mir-3/–309 lines and Dr. Stephen Cohen for the *mir-309-C* mutant. We thank the modENCODE consortium, the Trudy FC Mackay and Sam Griffiths-Jones labs for their genomics data, and Drosophila Genomics Resource Center (supported by NIH grant

2P40OD010949) for BG3-C2 and Kc167 cells. We are grateful to Ramanuj DasGupta for supporting AS in his lab. The miRNA overexpression study was initiated in Eric C Lai's lab at Sloan-Kettering Institute and was supported by the National Institutes of Health R01-GM083300 and R01-NS083833 to ECL. The authors appreciate generous support by ECL. Research in KO's group was supported by the National Research Foundation, Prime Minister's Office, Singapore under its NRF Fellowship Programme (NRF2011NRF-NRFF001-042) and Temasek Life Sciences Laboratory core funding. Research in GT-K's group was supported by NUS Faculty of Science startup grant R-154-000-536-133, AcRF grant R-154-000-562-112, and Lee Hiok Kwee fund grant R-154-000-582-651. Work in NST's laboratory was supported by an Academic Research Fund (AcRF) grant (MOE2014-T2-2-039). The content is solely the responsibility of the authors and does not necessarily represent the official views of these agencies.

## Additional information

### Funding

| Funder | Grant reference number | Author |
| --- | --- | --- |
| National Research Foundation Singapore | NRF2011NRF-NRFF001-042 | Li Zhou<br>Mandy Yu Theng Lim<br>Katsutomo Okamura |
| National Institutes of Health | R01-GM083300 | Diane Bortolamiol-Becet |
| Ministry of Education - Singapore | MOE2014-T2-2-039 | Nicholas Tolwinski |
| National University of Singapore | R-154-000-536-133 | Greg Tucker-Kellogg |
| National Institutes of Health | R01-NS083833 | Diane Bortolamiol-Becet |
| National University of Singapore | R-154-000-562-112 | Greg Tucker-Kellogg |
| National University of Singapore | R-154-000-582-651 | Greg Tucker-Kellogg |

The funders had no role in study design, data collection and interpretation, or the decision to submit the work for publication.

### Author contributions

Li Zhou, Mandy Yu Theng Lim, Conceptualization, Resources, Data curation, Formal analysis, Investigation, Visualization, Methodology, Writing—review and editing; Prameet Kaur, Diane Bortolamiol-Becet, Resources, Data curation, Formal analysis, Writing—review and editing; Abil Saj, Conceptualization, Data curation, Supervision, Writing—review and editing; Vikneswaran Gopal, Data curation, Formal analysis, Methodology, Writing—review and editing; Nicholas Tolwinski, Resources, Data curation, Formal analysis, Supervision, Funding acquisition, Writing—review and editing; Greg Tucker-Kellogg, Conceptualization, Data curation, Formal analysis, Supervision, Funding acquisition, Writing—review and editing; Katsutomo Okamura, Conceptualization, Resources, Data curation, Formal analysis, Supervision, Funding acquisition, Visualization, Writing—original draft

### Author ORCIDs

Katsutomo Okamura (iD) http://orcid.org/0000-0002-8316-0960

### Decision letter and Author response

Decision letter https://doi.org/10.7554/eLife.38389.047
Author response https://doi.org/10.7554/eLife.38389.048

# Additional files

## Supplementary files

• Supplementary file 1. Library statistics (related to *Figure 1*). Sheet 1: small RNA library mapping to dm3 genome. Sheet 2: small RNA library category mapping statistics. Sheet 3: small RNA library spike-in counts. Sheet 4: public total RNA-seq library used in this study. Sheet 5: public small RNA-seq library used in this study.
DOI: https://doi.org/10.7554/eLife.38389.027

• Supplementary file 2. miRNA expression level (related to *Figure 1*). Sheet 1: miRNA Reads Counts (normalized by the number of genomic locations). Sheet 2: miRNA normalized reads (RPTS).
DOI: https://doi.org/10.7554/eLife.38389.028

• Supplementary file 3. Pri-miRNA transcription activity (related to *Figure 1*). Sheet 1: Density of total RNA-seq reads in the upstream region of miRNA hairpin.
DOI: https://doi.org/10.7554/eLife.38389.029

• Supplementary file 4. Summary of multiple regression analysis (related to *Figures 1* and *2*).
DOI: https://doi.org/10.7554/eLife.38389.030

• Supplementary file 5. ANOVA analysis summary (related to *Figure 3*).
DOI: https://doi.org/10.7554/eLife.38389.031

• Supplementary file 6. DGRP polymorphic target analysis (related to *Figure 7*). Sheet 1: Polymorphic target sites and their allele frequencies in the DGRP dataset. To draw the chart in *Figure 7—figure supplement 1*, polymorphic target sites were binned based on the allele frequency values (highlighted in red). Sheet 2: Counts of polymorphic target sites in each bin in *Figure 7—figure supplement 1* charts. The fraction of polymorphic target sites ('fraction': highlighted in red) was used for the chart. Sheet 3: Derived target sites and their allele frequencies in the DGRP dataset. To draw the chart in *Figure 7G*, derived target sites were binned based on the derived allele frequency values ('DAF': highlighted in red). Sheet 4: Counts of derived target sites in each bin in *Figure 7F* charts. The fraction of derived target sites ('fraction': highlighted in red) was used for the chart.
DOI: https://doi.org/10.7554/eLife.38389.032

• Supplementary file 7. Oligos used in this study (related to Materials and methods).
DOI: https://doi.org/10.7554/eLife.38389.033

• Supplementary file 8. Genomic coordinates of additional isoforms of miRNA host transcripts (related to Materials and methods).
DOI: https://doi.org/10.7554/eLife.38389.034

• Transparent reporting form
DOI: https://doi.org/10.7554/eLife.38389.035

## Data availability

The small RNA library data produced for this study are deposited at NCBI SRA under SRP109269.

The following dataset was generated:

| Author(s) | Year | Dataset title | Dataset URL | Database, license, and accessibility information |
|---|---|---|---|---|
| Temasek life sciences laboratory | 2017 | Integrated profiling of mature and primary miRNAs reveals the importance of miRNA stability and alternative selection of primary miRNA isoforms | https://trace.ncbi.nlm.nih.gov/Traces/sra/?study=SRP109269 | Publicly available at the NCBI Gene Expression Omnibus (accession no: SRP109269) |

The following previously published datasets were used:

| Author(s) | Year | Dataset title | Dataset URL | Database, license, and accessibility information |
|---|---|---|---|---|
| Berkeley Drosophila Genome Project (BDGP) | 2014 | D. melanogaster Total RNA-Seq, ChIP-seq | https://trace.ncbi.nlm.nih.gov/Traces/sra/?study=SRP001696 | Publicly available at the NCBI Gene Expression Omnibus |

| | | | | |
|---|---|---|---|---|
| | | | | (accession no: SRP001696) |
| Ninova M | 2014 | Small RNA expression throughout the development of Drosophila virilis | https://www.ncbi.nlm.nih.gov/geo/query/acc.cgi?acc=GSE54009 | Publicly available at the NCBI Gene Expression Omnibus (accession no: GSE54009) |
| Mackay TFC | 2014 | Drosophila genetics reference panel 2 | http://dgrp2.gnets.ncsu.edu/ | Publicly available at the dataset URL (VCF file for the DGRP Freeze 2.0 calls) |
| White KP | 2009 | Genome-wide maps of chromatin state in staged Drosophila embryos, ChIP-seq | https://trace.ncbi.nlm.nih.gov/Traces/sra/?study=SRP001424 | Publicly available at the NCBI Gene Expression Omnibus (accession no: SRR030329) |

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
