## [Decision Letter]

[Editors’ note: a previous version of this study was rejected after peer review, but the authors submitted for reconsideration. The first decision letter after peer review is shown below.]

Thank you for submitting your work entitled "Importance of miRNA stability and alternative primary miRNA isoforms in gene regulation during *Drosophila* development" for consideration by *eLife*. Your article has been reviewed by a three reviewers, and the evaluation has been overseen by Timothy Nilsen (Reviewing Editor) and a Senior Editor. The following individuals involved in review of your submission have agreed to reveal their identity: Hervé Seitz (Reviewer #2).

Our decision has been reached after consultation between the reviewers. Based on these discussions and the individual reviews below, we regret to inform you that your work will not be considered further for publication in *eLife*.

The reviewers felt that the manuscript was potentially quite interesting, but each raised a number of concerns, some of which need to be addressed via further experiments. We are declining the manuscript but encourage resubmission if and when these issues have been thoroughly addressed.

Reviewer #1:

Importance of miRNA stability and alternative primary miRNA isoforms in gene regulation during *Drosophila* development.

In this study, the authors have used miRNA sequencing of different *Drosophila* developmental stages. Generally, they find global changes during embryogenesis and miRNAs can be grouped according to their expression levels. The sequencing data is confirmed by Northern blotting of a number of selected miRNAs. In order to compare miRNA abundance with transcription rates, the authors analyzed publicly available total RNAseq data from different *Drosophila* developmental stages. Overall, transcription levels correlate with the abundance of mature miRNAs and thus they conclude that transcription is one important aspect of shaping miRNA profiles. Closer examination of cluster miRNAs reveals that these miRNAs are not equally expressed. In fact, the authors identify different isoforms of pri-miRNAs with different TSS or TTS explaining differential expression of several of the cluster miRNAs. Another important aspect of miRNA profile determination is miRNA stability and turn over. As an example, they analyzed the miR-309 cluster, which is transcribed at early stages. Nevertheless, some of the mature miRNAs are still present at late stages while others are rapidly degraded. Genome-wide analyses revealed that the presence of a 5' U is a major determinant of miRNA stability. Finally, they validate Vang as a target of miR-3 and miR-309 and show that regulation of miRNA stability is functionally important for embryogenesis.

The authors present a comprehensive analysis of features that determine mature miRNA expression profiles during different stages of *Drosophila* embryogenesis. The manuscript is well written, and the results are presented clearly. Several of the findings are not entirely new but are combined into a comprehensive analysis. Nevertheless, there are a number of points that need clarification.

1) Figure 4C: The authors claim in the text that all three miRNAs are equally degraded in the BG3-c2 cell line. This is not obvious from Figure 4C. There is not much difference between of miR-277 between the left (Kc167) and the right panel (BG3-c2). Maybe it would be clearer if the signals of the Northern blots were quantified.

2) One main claim of the study is that uridines (Us) at the 5' end determine the half-lives of miRNAs. The authors mutate a miRNA that contains an A at the 5' position to a U and find that this change mildly stabilizes the miRNA. This is not new since structural work has demonstrated that Us are better bound by the MID domain than other nucleotides. The authors should analyze these finding more systematically. They should mutate miR-283 to all four nucleotides at the 5' end and analyze decay rates. Which of the miRNAs shown in Figure 5 contain Us at the 5' end? Is this consistent with the overall finding that Us increase stability?

3) The authors should also test whether the dwelling time on AGO1 is changed when Us or other terminal nucleotides are present at the 5' end. This would mechanistically explain the observed effects.

4) It has been reported that primary transcripts of clustered miRNAs can fold into specific structures with limited Drosha accessibility for some of the miRNAs. Is folding relevant here as well? Folding could also be dynamic allowing for differential Drosha processing during *Drosophila* embryogenesis.

5) It has been shown recently that specific human primary miRNAs can be cleaved into two intermediate pri-miRNAs independently of the microprocessor (Du et al., 2015). Since the authors only look at transcript levels, shorter variants could also derive from such cleavage events rather than differential TSS or TTS.

6) The RNAseq data that is used to model transcription rates only detects primary transcripts. It could be possible that pre-miRNAs are not cloned and sequenced in these data sets because adaptor ligation might be difficult or at least inefficient for double-stranded RNAs. Thus, pre-miRNA molecules would not be present in the entire analysis. Some of the conclusions might be difficult without knowing anything about pre-miRNA levels. The authors should check whether this is relevant for the conclusions that have been made in this manuscript.

Reviewer #2:

In this manuscript, Zhou and colleagues use a replicated Small RNA-Seq time-course analysis of small RNAs in *Drosophila* embryos to measure the dynamics of miRNA abundance during embryonic development. Comparing their small RNA data to published RNA-seq libraries (which can inform them on the abundance of pri-miRNAs), they can then use a simplified mathematical model to describe steady-state miRNA accumulation by a first-order processing of pri-miRNAs, and a first-order decay of mature miRNAs. The authors conclude that, for most miRNAs, both pri-miRNA processing and mature miRNA decay are rather constant during embryogenesis (meaning that mature miRNA accumulation is mostly governed by the pri-miRNA transcription rate), but that modeling analysis is very poorly detailed, and the accuracy of its results appears fragile (see below). The authors then perform a comparison of the dynamics of miRNAs expressed from clustered genes (which are often assumed to be co-transcribed) and propose explanations for the reasons why the expression of clustered miRNAs can appear uncoordinated (dynamic regulation of alternative transcription initiation and termination; differential stability of the mature miRNA). They finally interrogate the biological functionality of such differential miRNA stability, but the conclusions are not very clear (see below).

Overall, this could be a very interesting article (not much is known about the control of the kinetics of miRNA expression in vivo), but several important issues have to be solved.

Essential revisions:

1) Normalization is always an issue in RNA-Seq, and the authors are rightfully concerned about it. But their claim that spike-in normalization is "robust and reliable" (subsection “Global changes in the bulk miRNA abundance during embryogenesis) is not rigorously supported by the data. It is important to realize that spike-in normalization is formally equivalent to a normalization to the amount of total RNA. When spike-ins are introduced in the RNA sample prior to library preparation, they are introduced in proportion to the quantity of total RNA (e.g., X fmol of oligos per microgram of RNA). Normalizing to spike-ins is thus equivalent to normalizing to total RNAs or (almost equivalently) to full-length ribosomal RNAs (because they constitute most of the cellular pool of RNA). That normalization scheme does not account for potential changes in total intracellular RNA (e.g., after the onset of zygotic transcription). Please note, too, that the Northern blots shown on Figure 2 were loaded with a fixed amount of total RNA (10 μg RNA per lane) so it is no surprise that spike-in-normalized Small RNA-Seq and Northern blots are in good agreement. I would recommend defining precisely what are the expected features of a "robust and reliable" normalization (should the values be proportional to the intracellular concentration of the RNA of interest? to its fraction in total RNA? to its fraction in the small RNA population? etc.) before concluding anything about the robustness and reliability of this particular normalization scheme. As a matter of fact, I do think that spike-in normalization is good (i.e.: it makes sense to quantify RNAs by their fraction in total RNA), but this has to be explained explicitly, and Northern blots loaded with a fixed amount of total RNA cannot be seen as an independent validation.

2) The so-called "3D linear model" is very imprecisely explained. At the very least, the equation of the model should be presented, and the meaning of these mysterious symbols in Supplementary PDF 1 should be explained (what are the red crosses, squares, and circles? What is the black grid?). I am assuming that "upstream density" is the title of the y-axis (but please write it parallel to the axis, and centered on the axis); if it is not, then I am completely lost. My understanding of the strange figures shown on Supplementary PDF 1 is that the authors are representing graphically a linear relationship with two variables (changes in steady state miRNA level as a function of the initial miRNA level and of the density of pri-miRNA reads, which is assumed to be proportional to pri-miRNA abundance). The authors expect a linear relationship because pri-miRNA processing and mature miRNA decay are assumed to be first-order reactions. Hence, on these tridimensional plots shown in Supplementary PDF 1, the possible solutions of the equation would fall on a plane. Maybe the black grids represent these expected planes, but then I would expect a negative slope for the variable "initial miRNA level" (and a positive slope for "upstream density"). For many miRNAs (see bantam, miR-, miR-14 …), the slope for "initial miRNA level" is positive. It is possible that I completely misunderstood that whole analysis (the absence of details in the manuscript didn't help); or there is clearly something that needs to be discussed by the authors. One obvious possibility is that RNA-Seq measurements in embryos may not be precise enough for such an analysis, and these slopes are just heavily contaminated by some technical noise (which could even turn them positive). But if precision really is so bad, then there's not much to be concluded from that analysis (the main text should then be modified accordingly (subsection “Transcription levels of individual pri-miRNAs estimated by total RNA-seq analysis” and subsection “Genome-wide analysis of miRNA stability”).

3) The functional assessment of the destabilization of miR-3 (subsection “Biological importance of miR-3/-309 family miRNAs”) is not fully convincing. The authors want to know whether the regulated decay of miR-3 in late embryos triggers some phenotypes. But they assess it by over-expressing miR-3/-309 under an eq-gal4 driver, and nothing indicates that the magnitude of the resulting over-expression is the same than what would be observed in the absence of a regulated miRNA decay (which could be approximated by simply looking at the miRNA level in earlier embryos, and assuming it would remain unchanged in the absence of a regulated decay). Here, the observed bristle phenotype may simply be due to an exceedingly large, non-physiological over-expression of the miRNAs. For that experiment, it is important to measure how much the miRNAs have been over-expressed under the eq-gal4 driver (and use a weaker driver if they were too strongly over-expressed). I also have to report that the identification of Vang as a relevant target is somewhat deceiving. It is merely based on the fact that Vang is a TargetScan-predicted target, with a known role in the PCP pathway. But the luciferase assay is hardly meaningful (co-expressing an artificial reporter with an over-expressed *Drosophila* miRNA in human cells); it does not add much to the simple fact that Vang is a TargetScan-predicted target (once we know the 3´ UTR has a perfect seed match to the miRNA, it is quite obvious that an artificial co-expression of both would result in miRNA binding). The in vivo test (quantifying Vang expression in mir-309 cluster mutant embryos, over-expressing embryos, and wt embryos) is less artificial, but the results were also expected, given that Vang is a TargetScan-predicted target. A real, convincing assessment of the role of the miR-3/Vang interaction in the bristle phenotype would consist in the mutagenesis of the miR-3 binding sites in the Vang UTR in vivo, followed by an analysis of the bristle phenotype. This is quite some work, I am not sure it would fall in the scope of this manuscript, but in the absence of such an analysis, the whole Vang story appears a bit gratuitous and unjustified.

Reviewer #3:

The manuscript by Zhou et al., describes how the expression of miRNAs are regulated and they chose *Drosophila* embryogenesis as a system. They obtained miRNA expression profiles of embryo at different time windows by their own sequencing, and also obtained pri-miRNA profiles from modENCODE project. Their miRNA sequence analyses revealed several different patterns of changes in expression profiles through embryogenesis. Their pri-miRNA analysis suggested that transcription levels of miRNA gene (levels of pri-miRNAs) are involved in determination of miRNA expression levels. In addition to that, the authors raised regulated expression of precursor RNA isoforms and miRNA stability (degradation rate) as additional mechanism for regulation of miRNA expression levels, and finally, they showed functional significance of the regulated miRNA expression. The manuscript contains interesting insights into miRNA biology, and the following points would improve the manuscript.

1) There are some inconsistencies between Figure 5D and 5C. For example, miR-3 "input" bands look similar between 2-4h and 4-6h in Figure 5D, but their intensities are very different in Figure 5C. Northern blots for 2-4h and 4-6h in Figure 5D should be done in the same membrane to accurately compare.

2) In Figure 5D, "input" bands between 2-4h and 4-6h looked similar, but the amounts of each miRNA loaded onto AGO1 were very different (much more abundant in 2-4h than in 4-6h). Although the authors focused on "reduction rate", can’t the results suggest different efficiency of AGO1-loading? Also, purified AGO1 protein levels in the immunoprecipitates should be examined by Western blot.

3) In Figure 6A, I am curious how the other three nucleotides (5'A, 5'G, and 5'C) affect miRNA levels. It would be better not to collectively show "other 5'-nucleotide" but to show the data for each four nucleotides.

4) In Figure 6B, only one example (analyses of only miR-283) is not enough to say the effect of 5'-U in miRNA stability, because the expression of miRNA can cause many things in the cells. I would suggest that the analyses of at least 3-5 different miRNAs are required to confirm their hypothesis.

5) Figure 7 suggested regulation of Vang mRNA by miR-3/309. Because these miRNAs were drastically decreased through embryogenesis (Figure 3), it would be great if the authors quantify Vang mRNA levels through embryogenesis and observe anti-correlation of their expression profiles to further confirm that Vang mRNA is regulated by the miRNAs.

6) AGO1 protein level was not investigated, but it should be a factor involving in miRNA expression levels and stabilities. It would be great if AGO1 Western blot data at various time windows of *Drosophila* embryos were added.

[Editors’ note: what now follows is the decision letter after the authors submitted for further consideration.]

Thank you for submitting your article "Importance of miRNA stability and alternative primary miRNA isoforms in gene regulation during *Drosophila* development" for consideration by *eLife*. Your article has been reviewed by Timothy Nilsen as Reviewing Editor and James Manley as the Senior Editor and three reviewers. The following individuals involved in review of your submission have agreed to reveal their identity: Hervé Seitz (Reviewer #1).

The reviewers have discussed the reviews with one another and the Reviewing Editor has drafted this decision to help you prepare a revised submission.

This resubmitted manuscript is a significant improvement over the previous submission. The only remaining issue is the presentation of the 3D lineal model. Because this model is only presented in supplementary material, it is strongly suggested that the authors remove it from the manuscript. If they do so, the manuscript would be acceptable for *eLife*.

Reviewer #1:

In this new submission of a previously-reviewed manuscript, L. Zhou and colleagues quantify mature miRNAs and pri-miRNA precursors during *Drosophila* development, to derive molecular rules governing miRNA abundance in that dynamic process. The modified manuscript addresses most of my previous concerns, and it has been clearly improved. One of my main concerns remains though: the provided data suggest that the "3D lineal model" predicts mature miRNA abundance very poorly, most probably because of quantification inaccuracies (which may very well be impossible to improve with the current technologies). A precise description of that model is (still) desperately needed in the main text, as well as an estimation of the fitted coefficients' precision. The provided data suggest that precision is too low to permit the analyses that the authors perform – if so, then I would simply suggest to remove that modeling analysis (it would be mostly describing technical noise).

Once that issue is fixed, I think the manuscript will be a very good candidate for publication in *eLife*.

Essential revisions:

1) Regarding the "3D linear model" whose results are presented in Supplementary PDF 1: I appreciate the authors' clarification, but they mostly appear in their point-by-point response to the referees, and the readers themselves are left without much information. As I said in my evaluation of the first submission, it is important to write down the model's equation (please do that in the main text and write explicitly that the coefficient for "initial miRNA level" is expected to be negative, while the coefficient for "upstream density" is expected to be positive – and explain why, in each case). Without this, the reader will be completely lost. As for the accuracy of the estimation of coefficients by this model: the added sentence ("However, we note that this analysis has a caveat […]") is an improvement, but it lacks rigor. The fact that not every point falls on the plane is not a surprise (nothing is perfect in experimental science); the fact that they fall "far away" from the plane would be a problem, but for this we need to know how far is "far away". The authors' current formulation is deceiving ("significant" has a very precise meaning: it means that a statistical test was performed, and the null hypothesis was rejected with a given p-value cutoff). Here it looks like the authors use "significant" as a synonym for "large" (without defining how large is "large"), which it is not. Even more problematic, the fact that many slopes for "initial miRNA level" appear to be positive is not discussed. Honestly, from what I see, it looks like these measurements are extremely imprecise, and they cannot be used to conclude anything regarding the miRNA turnover rate. The fact that "5´ U" values end up begin mostly larger than "other 5´ nucleotide" values seems to be pure luck (with another, similarly imprecise RNA-Seq-based quantification, the values may have very well fallen in the opposite order). In their Point-by-point response, the authors say that the global analysis of every miRNA is certainly more precise than each individual measurement, which is certainly true – but whether this improvement is sufficient in order to perform an analysis like the one shown in Figure 6A remains to be demonstrated. For this, the authors need to evaluate the accuracy of their modeled coefficients (a linear modeling analysis does not only compute the fitted coefficients, it also provides an estimate of its precision; simply giving the coefficient without giving the precision is pointless). Given the aspect of the 3D graphs and given the dispersion of the points on Figure 6A (with the median for "5´ U" values being positive while their mean is negative), I expect precision to be too bad to allow a meaningful comparison of 5´ U miRNAs and 5´ non-U miRNAs (note that Figure 1D does not show that the model "worked reasonably well": most points fall next to the origin, with a high relative error). If indeed it is the case, then I recommend dropping the "3D linear model" analysis (which would be too noisy to be informative), and simply find another way to explain why the authors wanted to perform the analyses shown in Figure 6. (on a side note: Figure 6—figure supplement 1 is useless: if the authors wanted to compare 5´ U miRNAs to 5´ non-U miRNAs, they just had to compare the two datasets shown in Figure 6A, and they did not have to stratify "5´ non U" into "5´ A", "5´ C" and "5´ G" as shown in that supplement).

Reviewer #2:

The authors responded well to the raised points in the first review with additional experimental results. I believe the current version of the manuscript is suitable for publication.

Reviewer #3:

In this revised version of their manuscript, the authors have adequately addressed all points that I had raised on the previous version. They have added extra experimental data and discussed it appropriately. I am satisfied with the response to my comments.

---

## [Author Response]

[Editors’ note: the author responses to the first round of peer review follow.]

The reviewers felt that the manuscript was potentially quite interesting, but each raised a number of concerns, some of which need to be addressed via further experiments. We are declining the manuscript but encourage resubmission if and when these issues have been thoroughly addressed.Reviewer #1:.[…] 1) Figure 4C: The authors claim in the text that all three miRNAs are equally degraded in the BG3-c2 cell line. This is not obvious from Figure 4C. There is not much difference between of miR-277 between the left (Kc167) and the right panel (BG3-c2). Maybe it would be clearer if the signals of the Northern blots were quantified.

The experiments were repeated three times and the quantification results have been added as Figure 4—figure supplement 3. This analysis supported our conclusion that miR-277 expression was better correlated with miR-34 and miR-317 in BG3-c2 cells than in Kc167 and S2-R+ cells.

*2) One main claim of the study is that* uridines (Us) *at the 5' end determine the half-lives of miRNAs. The authors mutate a miRNA that contains an A at the 5' position to a U and find that this change mildly stabilizes the miRNA. This is not new since structural work has demonstrated that Us are better bound by the MID domain than other nucleotides. The authors should analyze these finding more systematically. They should mutate miR-283 to all four nucleotides at the 5' end and analyze decay rates. Which of the miRNAs shown in Figure 5 contain Us at the 5' end? Is this consistent with the overall finding that Us increase stability?*

To clarify, we do not believe the 5’ nucleotide is the sole determinant of miRNA stability. Based on our observations, we concluded that the 5’ nucleotides can influence miRNA stability and would be among a number of factors that determine half-lives. Therefore, if we compare the groups of miRNAs according to the 5’ nucleotides, we observe some trends as seen in Figure 6A. If we compare half-lives of a small number of miRNAs individually, the trend may not be obvious. In fact, if we only look at the mir-309 cluster miRNAs, the trend is not clear (Figure 5, Mir-309: G, Mir-3: U, Mir-286: U, Mir4: A, Mir-5: U, Mir-6: U). In particular, we believe that the instability of mir-3/309 is mediated by other mechanisms because the accelerated degradation rate cannot be explained by the effect of 5’ nucleotides that can only cause mild changes of stability as shown by the targeted mutagenesis (~20%, Figure 6D).

To further support our conclusion regarding the effects of miRNA 5’ nucleotides on mature miRNA stability, we made additional mutants by changing the 5’ end to all three possible nucleotides. This experiment is not as simple as it may sound. We needed to find miRNAs whose strand selection and the cleavage sites by Drosha/Dicer were not altered by the 5’ nucleotide changes. Through careful selection of miRNAs and experimental verification, we managed to find two additional miRNAs that were suitable for this analysis, and almost all possible mutants could be made (except the AtoG mutant of miR-263a, whose Drosha cleavage site was altered by the mutation). Unexpectedly, we observed stabilizing effects of 5’-C, in addition to the expected effect by 5’-U mutations. We added these results as Figure 6D and Figure 6—figure supplement 2. Taken together, we report a role for 5’ nucleotides in determining the mature miRNA half-lives.

3) The authors should also test whether the dwelling time on AGO1 is changed when Us or other terminal nucleotides are present at the 5' end. This would mechanistically explain the observed effects.

To our best knowledge, *Drosophila* AGO1 has not been used for single molecule imaging analysis potentially due to the low abundance of empty AGO1 in the fly system. The rapid degradation of empty AGO1 may be the cause of this (Smibert et al., 2013). The detailed single-molecule analysis is certainly an interesting direction, but precise mechanisms and dynamics of Argonaute-guide RNA interactions are beyond the scope of this manuscript. We plan to follow up this mechanism in the future as better tools are developed.

4) It has been reported that primary transcripts of clustered miRNAs can fold into specific structures with limited Drosha accessibility for some of the miRNAs. Is folding relevant here as well? Folding could also be dynamic allowing for differential Drosha processing during Drosophila embryogenesis.

It has been reported that the mammalian mir-17-92 cluster transcript is folded into globular structures, which reduces processing efficiency of miRNA hairpins that are located in the core of the structure (Chaulk et al., 2014; Chaulk et al., 2017). As the same reviewer points out below, the same cluster has been reported to undergo post-transcriptional regulation via the action of the endonuclease CPSF3 to cleave the pri-miRNA to generate “pro-miRNA” in which all hairpins in this cluster can be efficiently processed by Drosha (Du et al., 2015). There is certainly additional complexity observed with processing of clustered miRNAs as also highlighted by recent reports. Therefore, we added more information in the Discussion section to highlight these studies further. We believe that the disparity we detected with expression changes between members of certain clusters may reflect some aspects of complex post-transcriptional regulation of clustered miRNA processing.

5) It has been shown recently that specific human primary miRNAs can be cleaved into two intermediate pri-miRNAs independently of the microprocessor (Du et al. Cell, 2015). Since the authors only look at transcript levels, shorter variants could also derive from such cleavage events rather than differential TSS or TTS.

While we do not exclude this possibility pointed out by the reviewer, we found that histone modification marks support the hypothesis of alternative TSSs in this cluster (New Figure 4—figure supplement 2). We added this information and mentioned both possibilities in the main text.

6) The RNAseq data that is used to model transcription rates only detects primary transcripts. It could be possible that pre-miRNAs are not cloned and sequenced in these data sets because adaptor ligation might be difficult or at least inefficient for double-stranded RNAs. Thus, pre-miRNA molecules would not be present in the entire analysis. Some of the conclusions might be difficult without knowing anything about pre-miRNA levels. The authors should check whether this is relevant for the conclusions that have been made in this manuscript.

We agree with the reviewer that the total RNA libraries (that are generally selective for >200nt species) are devoid of pre-miRNAs. While there are protocols for pre-miRNA expression profiling, those experiments are not straightforward and accurately measuring pre-miRNA levels remains a challenge. Therefore, we decided to make conclusions based on the levels of pri-miRNAs and cognate mature miRNAs. We believe that the lack of pre-miRNA information does not affect our conclusions (Usage of TSS/TTS, the stability of mir-309 cluster miRNAs and roles of 5’ nucleotide in loaded mature miRNAs).

Reviewer #2:[…] 1) Normalization is always an issue in RNA-Seq, and the authors are rightfully concerned about it. But their claim that spike-in normalization is "robust and reliable" (p. 6) is not rigorously supported by the data. It is important to realize that spike-in normalization is formally equivalent to a normalization to the amount of total RNA. When spike-ins are introduced in the RNA sample prior to library preparation, they are introduced in proportion to the quantity of total RNA (e.g., X fmol of oligos per microgram of RNA). Normalizing to spike-ins is thus equivalent to normalizing to total RNAs or (almost equivalently) to full-length ribosomal RNAs (because they constitute most of the cellular pool of RNA). That normalization scheme does not account for potential changes in total intracellular RNA (e.g., after the onset of zygotic transcription). Please note, too, that the Northern blots shown on Figure 2 were loaded with a fixed amount of total RNA (10 μg RNA per lane) so it is no surprise that spike-in-normalized Small RNA-Seq and Northern blots are in good agreement. I would recommend defining precisely what are the expected features of a "robust and reliable" normalization (should the values be proportional to the intracellular concentration of the RNA of interest? to its fraction in total RNA? to its fraction in the small RNA population? etc.) before concluding anything about the robustness and reliability of this particular normalization scheme. As a matter of fact, I do think that spike-in normalization is good (i.e.: it makes sense to quantify RNAs by their fraction in total RNA), but this has to be explained explicitly, and Northern blots loaded with a fixed amount of total RNA cannot be seen as an independent validation.

The reviewer points out potential issues with normalization methods and interpretation of the results. We decided to use the normalization method that is based on the assumption that the amount of total RNA per embryo stays constant. However, we could not find a study explicitly testing the amount of total RNA per embryo in different stages of development. We attempted to estimate the amounts of total RNA per embryo at different time windows, and the results suggested that the total RNA per embryo stayed roughly at the same level. Therefore, we decided to adhere to the spikein-based normalization for miRNA analysis. We also clarified in the main text that Northern blotting is to verify the sequence-based quantification results that relies on the same assumption.

2) The so-called "3D linear model" is very imprecisely explained. At the very least, the equation of the model should be presented, and the meaning of these mysterious symbols in Supplementary PDF 1 should be explained (what are the red crosses, squares, and circles? what is the black grid?). I am assuming that "upstream density" is the title of the y-axis (but please write it parallel to the axis, and centered on the axis); if it is not, then I am completely lost. My understanding of the strange figures shown on Supplementary PDF 1 is that the authors are representing graphically a linear relationship with two variables (changes in steady state miRNA level as a function of the initial miRNA level and of the density of pri-miRNA reads, which is assumed to be proportional to pri-miRNA abundance). The authors expect a linear relationship because pri-miRNA processing and mature miRNA decay are assumed to be first-order reactions. Hence, on these tridimensional plots shown in Supplementary PDF 1, the possible solutions of the equation would fall on a plane. Maybe the black grids represent these expected planes, but then I would expect a negative slope for the variable "initial miRNA level" (and a positive slope for "upstream density"). For many miRNAs (see bantam, miR-, miR-14 …), the slope for "initial miRNA level" is positive. It is possible that I completely misunderstood that whole analysis (the absence of details in the manuscript didn't help); or there is clearly something that needs to be discussed by the authors. One obvious possibility is that RNA-Seq measurements in embryos may not be precise enough for such an analysis, and these slopes are just heavily contaminated by some technical noise (which could even turn them positive). But if really precision is so bad, then there's not much to be concluded from that analysis (the main text should then be modified accordingly subsection “Transcription levels of individual pri-miRNAs estimated by total RNA-seq analysis” and subsection “Genome-wide analysis of miRNA stability”).

We apologize for the poor presentation of the charts and inadequate descriptions in this file, and we appreciate the reviewer’s effort to correctly interpret the results. As the reviewer describes, we meant to present the linear relationship between the initial miRNA level, the density of pri-miRNA reads and the change rate of the mature miRNA. This equation worked reasonably well in predicting the overall change rate as shown in Figure 1D. However, the slope for “initial miRNA level” was variable and some genes even showed positive slope in contrast to our expectation, as correctly pointed out by the reviewer. We believe that this is due to the high variation of the estimated values and slow degradation of mature miRNA species. Therefore, we decided not to overly interpret the slope for each miRNA. However, when the values were considered in groups of larger number of genes, we believe the distributions of slope values carry meaningful information. For example, the means of slopes were negative values (mean_5’U_=-0.0016, mean_non-5’U_=-0.088). Although we recognize the fluctuation of the estimation of individual degradation rates, the accuracy was high enough to detect the difference in the mature miRNA stability of 5’U- and non-5’U-species (Figure 6A). This led to the finding that 5’ nucleotides may affect mature miRNA stability, and this hypothesis could be verified experimentally by using individual miRNA constructs mutating 5’ nucleotides (Figure 6B-D). Therefore, we believe that this analysis could provide meaningful information depending on how we interpret it. We modified the main text to clarify these points.

3) The functional assessment of the destabilization of miR-3 (subsection “Biological importance of miR-3/-309 family miRNAs”) is not fully convincing. The authors want to know whether the regulated decay of miR-3 in late embryos triggers some phenotypes. But they assess it by over-expressing miR-3/-309 under an eq-gal4 driver, and nothing indicates that the magnitude of the resulting over-expression is the same than what would be observed in the absence of a regulated miRNA decay (which could be approximated by simply looking at the miRNA level in earlier embryos, and assuming it would remain unchanged in the absence of a regulated decay). Here, the observed bristle phenotype may simply be due to an exceedingly large, non-physiological over-expression of the miRNAs. For that experiment, it is important to measure how much the miRNAs have been over-expressed under the eq-gal4 driver (and use a weaker driver if they were too strongly over-expressed). I also have to report that the identification of Vang as a relevant target is somewhat deceiving. It is merely based on the fact that Vang is a TargetScan-predicted target, with a known role in the PCP pathway. But the luciferase assay is hardly meaningful (co-expressing an artificial reporter with an over-expressed Drosophila miRNA in human cells); it does not add much to the simple fact that Vang is a TargetScan-predicted target (once we know the 3´ UTR has a perfect seed match to the miRNA, it is quite obvious that an artificial co-expression of both would result in miRNA binding). The in vivo test (quantifying Vang expression in mir-309 cluster mutant embryos, over-expressing embryos, and wt embryos) is less artificial, but the results were also expected, given that Vang is a TargetScan-predicted target. A real, convincing assessment of the role of the miR-3/Vang interaction in the bristle phenotype would consist in the mutagenesis of the miR-3 binding sites in the Vang UTR in vivo, followed by an analysis of the bristle phenotype. This is quite some work, I am not sure it would fall in the scope of this manuscript, but in the absence of such an analysis, the whole Vang story appears a bit gratuitous and unjustified.

We redid the sensor assays in S2 cells and obtained similar results (Figure 7D), supporting our conclusion. Our point is that rapid degradation of miR-3 may have biological significance, and regulation of Vang could be part of it as ectopic expression of miR-3 causes detectable phenotypes that are consistent with Vang misregulation (Figure 7B and C). We agree with the reviewer’s comments regarding the level of overexpression because, if the level of overexpressed miRNAs exceed the highest level of natural miR-3/-309 expression, the phenotypes we are seeing may simply be artifacts due to the unnatural levels of overexpression. To test this, we decided to focus on the embryonic phenotypes and used the RNA samples that were used for Vang mRNA quantification with mir-3/-309 overexpression (Figure 7E). The levels of overexpressed miR-3/-309 were compared with the levels of those miRNAs at the highest expression peaks (Figure 7—figure supplement 1). For this analysis, we only used embryos overexpressing miR-3/-309 by the ubiquitous da-Gal4 because it is difficult to measure the expression levels for miR-3/-309 driven by eq-Gal4, which activates transcription only in a part of the wing disc (Tang and Sun, 2002). We found that the levels of overexpressed miR-3/-309 driven by da-Gal4 reached only <50% of the levels of endogenous miR-3/-309 at 3.5-4h embryos. Because the mir-309 cluster is known to be transcribed broadly in the somatic cells (doi: 10.1073 pnas.0507817102, 10.1073 pnas.0508823102), we believe that our experiment mimics a situation where the miR-3/-309 mature products were not degraded quickly.

The strictest experiment to demonstrate this is to identify the elements that make miR-3/-309 unstable, generate a mutant that has the same target specificity as the wild-type miR-3/-309, and analyze the organismal phenotype. We have started experiments to identify the destabilizing elements of miR-3/-309 and successful identification of the mechanisms would allow us to design the stricter experiments to study biological effects of miR-3/-309 degradation. However, we believe that identification of mechanisms and generation of mutants are beyond the scope of the current manuscript. In the revised manuscript, we added the results testing the levels of overexpressed miR-3/-309 (Figure 7—figure supplement 1) and discussed this point further.

Reviewer #3:[…] 1) There are some inconsistencies between Figure 5D and 5C. For example, miR-3 "input" bands look similar between 2-4h and 4-6h in Figure 5D, but their intensities are very different in Figure 5C. Northern blots for 2-4h and 4-6h in Figure 5D should be done in the same membrane to accurately compare.

In our previous Figure 5D, we did not adjust figures to show the differences between 2-4h and 4-6h input lanes, because the main point was to show that the loading efficiencies were similar between mir-309 cluster members. In the new manuscript, we added quantified results and Figure 5D and E charts show similar trends between the two time windows: decrease of miR-3/-309 and increase of the other miRNAs from the cluster.

2) In Figure 5D, "input" bands between 2-4h and 4-6h looked similar, but the amounts of each miRNA loaded onto AGO1 were very different (much more abundant in 2-4h than in 4-6h). Although the authors focused on "reduction rate", can’t the results suggest different efficiency of AGO1-loading? Also, purified AGO1 protein levels in the immunoprecipitates should be examined by Western blot.

The difference in the original result stems from the difference in experimental conditions including the concentrations of the lysate and the amounts of antibody used. We repeated the experiment with a standardized experimental condition. To provide more quantitative information, we added normalized quantified values (Signal intensity was normalized to the intensity of the signal in the input lane of 2–4 hour samples for each miRNA species). We also added Western blotting results as Figure 5—figure supplement 2.

3) In Figure 6A, I am curious how the other three nucleotides (5'A, 5'G, and 5'C) affect miRNA levels. It would be better not to collectively show "other 5'-nucleotide" but to show the data for each four nucleotides.

We added the figure as Figure 6—figure supplement 1. The three groups of miRNAs that had other 5’ nucleotides (5'A, 5'G, and 5'C) showed generally lower values compared with those for the 5’U species. However, due to the small number of miRNAs that satisfied expression criteria (3-9 miRNA species could be used for each 5’-nucleotide group), we did not perform statistical tests as we felt that it is difficult to make a meaningful conclusion with such small numbers of genes.

4) In Figure 6B, only one example (analyses of only miR-283) is not enough to say the effect of 5'-U in miRNA stability, because the expression of miRNA can cause many things in the cells. I would suggest that the analyses of at least 3-5 different miRNAs are required to confirm their hypothesis.

Similar comments were made by reviewer 1 as well. To further support our conclusion regarding the effects of miRNA 5’ nucleotides on mature miRNA stability, we made additional mutants by changing the 5’ end to all three possible nucleotides. This experiment is not as simple as it may sound. We needed to find miRNAs whose strand-selection and the cleavage sites by Drosha/Dicer were not altered by the 5’ nucleotide changes. Through careful selection of miRNAs and experimental verification, we managed to find two additional miRNAs that were suitable for this analysis, and almost all possible mutants could be made (except the A-to-G mutant of miR-263a, whose Drosha cleavage site was altered by the mutation). Unexpectedly, we observed stabilizing effects of 5’-C, in addition to the expected effect by 5’-U mutations. We added these results as Figure 6D and Figure 6—figure supplement 2. Taken together, we report a role for 5’ nucleotides in determining the mature miRNA half-lives.

5) Figure 7 suggested regulation of Vang mRNA by miR-3/309. Because these miRNAs were drastically decreased through embryogenesis (Figure 3), it would be great if the authors quantify Vang mRNA levels through embryogenesis and observe anti-correlation of their expression profiles to further confirm that Vang mRNA is regulated by the miRNAs.

We added this analysis as Figure 7—figure supplement 2. Indeed, the decrease of miR-3/-309 preceded the increase of Vang mRNA.

6) AGO1 protein level was not investigated, but it should be a factor involving in miRNA expression levels and stabilities. It would be great if AGO1 Western blot data at various time windows of Drosophila embryos were added.

The results are added as Figure 1—figure supplement 3. We observed a correlation between the AGO1 protein level and the total miRNA levels. However, because the mutual dependence of AGO1 and mature miRNAs for their stability (Smibert et al., 2013), this result itself did not allow us to conclude whether the level of AGO1 protein is limiting the amount of bulk mature miRNAs. This has to be further studied perhaps when researchers find the mechanism of empty AGO1 degradation and generate mutants of AGO1 whose stability does not rely on loaded guide RNAs. We discussed the Western blotting results in the main text.

[Editors' note: the author responses to the re-review follow.]

This resubmitted manuscript is a significant improvement over the previous submission. The only remaining issue is the presentation of the 3D lineal model. Because this model is only presented in supplementary material, it is strongly suggested that the authors remove it from the manuscript. If they do so, the manuscript would be acceptable for eLife.

Thank you for the opportunity to resubmit this revised version of the manuscript. As suggested by the editors, we removed the Supplementary PDF, which the reviewers and editors appeared to find confusing rather than useful. We also changed the order figures for a better flow. In addition, we made changes to address specific points that were raised by reviewer 1.

Reviewer #1:[…] 1) Regarding the "3D linear model" whose results are presented in Supplementary PDF1: I appreciate the authors' clarification, but they mostly appear in their Point-by-point response to the referees, and the readers themselves are left without much information. Given the aspect of the 3D graphs and given the dispersion of the points on Figure 6A (with the median for "5´ U" values being positive while their mean is negative), I expect precision to be too bad to allow a meaningful comparison of 5´ U miRNAs and 5´ non-U miRNAs (note that Figure 1D does not show that the model "worked reasonably well": most points fall next to the origin, with a high relative error). If indeed it is the case, then I recommend dropping the "3D linear model" analysis (which would be too noisy to be informative), and simply find another way to explain why the authors wanted to perform the analyses shown in Figure 6. (on a side note: Figure 6—figure supplement 1 is useless: if the authors wanted to compare 5´ U miRNAs to 5´ non-U miRNAs, they just had to compare the two datasets shown in Figure 6A, and they did not have to stratify "5´ non U" into "5´ A", "5´ C" and "5´ G" as shown in that supplement).

As suggested by the reviewer, we edited the main text to more clearly explain the model and the assumptions used. The reviewer is also concerned about the inaccuracy of the results that were previously shown as Figure 6A to present the predicted relative degradation rates of 5’-U and non-5’-U species. Although it is not surprising that some coefficients show positive values when the true value is expected to be very close to zero (as mature miRNAs are generally stable) and the statistical test indicated a small p-value with a widely used method (p=0.01, Kolmogorov-Smirnov test) when the values for the 5’-U and non-5’-U groups were compared, we recognize the risk of using noisy data for making conclusions. According to the reviewer’s comments, we made it clear that this analysis did not allow us to make a confident conclusion regarding the miRNA degradation rates by explicitly explaining the observed fluctuations and highlighting the unexpected positive median value for the 5’-U group. However, we kept the chart (previously main Figure 6A) as a figure supplement (now Figure 3—figure supplement 1) to explain what prompted us to mutate the 5’ nucleotides of the model miRNAs. It was difficult to explain this in another way. We performed these mutagenesis experiments only because we observed the difference in the distribution of coefficients between the 5’-U and non-5’-U groups as described in the manuscript.